# Synoptic atmospheric circulation patterns associated with deep persistent slab avalanches in the western United States

Andrew R. Schauer[1], Jordy Hendrikx[1], Karl W. Birkeland[2,1], Cary J. Mock[3]

[1] Snow and Avalanche Lab, Department of Earth Sciences, Montana State University, P.O. Box 173480, Bozeman, MT, 59717, USA

[2] USDA Forest Service National Avalanche Center, P.O. Box 130, Bozeman, MT, 59771, USA

[3] Department of Geography, University of South Carolina, Columbia, SC, 29208-0001

*Correspondence to*: Andrew R. Schauer (andrew_schauer@hotmail.com)

**Abstract.** Deep persistent slab avalanches are capable of destroying infrastructure and are usually unsurvivable to those who are caught. Formation of a snowpack conducive to deep persistent slab avalanches is typically driven by meteorological conditions occurring in the beginning weeks to months of the winter season, and yet the avalanche event may not occur for several weeks to months later. While predicting the exact timing of the release of deep persistent slab avalanches is difficult, onset of avalanche activity is commonly preceded by rapid warming, heavy precipitation, or high winds. This work investigates the synoptic drivers of deep persistent slab avalanches at three sites in the Western USA with long records: Bridger Bowl, Montana; Jackson, Wyoming; and Mammoth Mountain, California. We use self-organizing maps to generate twenty synoptic types that summarize 5,899 daily 500 mb geopotential height maps for the winters (November – March) of 1979/80 – 2017/18. For each of the three locations, we identify major and minor deep persistent slab avalanche seasons, and analyze the number of days represented by each synoptic type during the beginning (November – January) of the major and minor seasons. We also examine the number of days assigned to each synoptic type during the 72 hours preceding deep persistent slab avalanche activity for both dry and wet slab events. Each of the three sites exhibits a unique distribution of the number of days assigned to each synoptic type during November – January of major and minor seasons, and for the 72-hour period preceding deep persistent slab avalanche activity. This work identifies the synoptic scale atmospheric circulation patterns contributing to deep persistent slab instabilities, and the patterns that commonly precede deep persistent slab avalanche activity. By identifying these patterns, we provide an improved understanding of deep persistent slab avalanches, and an additional tool to anticipate the timing of these difficult-to-predict events.

## 1 Introduction

Deep persistent slab avalanches are challenging to predict and generally more destructive than other snow avalanche types. They threaten infrastructure, transportation, and recreationists in snow-covered mountain regions around the world (McClung and Schaerer, 2006). Deep persistent slab avalanches occur due to two different mechanisms, depending on whether they are primarily dry or wet. Dry deep persistent slab avalanches occur when a weak layer in the snowpack fails due to applied stress from an external load such as new and wind-transported snow, cornice fall, explosives, or the weight of a human. In wet deep persistent slab avalanches, the weak layer typically fails due to the introduction of liquid water to the snowpack, which deteriorates the bonds between grains in the weak layer such that the weak layer can no longer support the weight of the overlying slab (Baggi and Schweizer, 2009; Marienthal et al., 2012; Pietzsch, 2009).

A snowpack conducive to deep persistent slab avalanches is characterized by two key features: (1) A persistent weak layer of snow that exists in the snowpack for days, weeks, or even months after it is buried, and (2) A thick, cohesive slab of snow overlying that persistent weak layer. Although there is no specific universal slab thickness threshold defining the difference between a persistent slab avalanche and a 'deep' persistent slab avalanche, previous research has investigated events with minimum crown depths ranging from 0.8 m to 1.2 m (Conlan and Jamieson, 2016; Conlan et al., 2014; Marienthal et al., 2015; Savage, 2006).

The relationship between slab depth and snow stability is complex. Although the crack length required for onset of fracture propagation within a weak layer tends to decrease with increasing slab thickness and density, crack length tends to increase with increasing slab stiffness (Gaume et al., 2017). Schweizer and Camponovo (2001) found that when a load is applied at the snow surface, the area affected by that load increases with depth, effectively decreasing the magnitude of stress with depth. In their experiments, stresses measured at one meter depth were equal to roughly 10% of the applied load at the surface. Additionally, as slab stiffness increases, stress is more effectively dispersed with depth, which results in an additional decrease in applied stress at the buried weak layer (Thumlert and Jamieson, 2014). The result of these processes is that as the depth to the weak layer increases, it becomes increasingly difficult to initiate failure in a weak layer, yet propagation often becomes more likely. This leads to large uncertainty concerning the exact timing of deep slab release, making these events particularly difficult to predict.

Formation of a snowpack conducive to deep persistent slab avalanches is typically controlled by meteorological conditions occurring during the early months of the winter season (Marienthal et al. 2015), with low snow accumulation and cold temperatures driving weak layer development near the ground. Onset of slab avalanche activity is commonly preceded by new or wind-transported snow loading, rapid warming, or rain-on-snow events (Conlan et al., 2014; Davis et al., 1999; Marienthal et al., 2012). It is often the case that a weak layer will facilitate widespread avalanche activity when it is initially buried, and transition to a period of dormancy where avalanches are less common as the thickness of the overlying slab increases (Statham et al., 2018). Such a snowpack may even undergo multiple loading events with little or no resulting avalanche activity before being pushed to its breaking point, resulting in a widespread avalanche cycle (e.g. Marienthal et al., 2012).

Associations between meteorological events and slab avalanche activity have long been used as a tool to aid in avalanche forecasting (e.g. Atwater, 1954; LaChapelle, 1966; LaChapelle, 1980), but extending this approach to persistent deep slab avalanches is not as well understood. Recent research has studied meteorological thresholds prior to the onset of avalanche activity, highlighting the importance of precipitation totals, daily maximum temperatures, and rain events in time periods ranging from 24 hours to 7 days prior to onset of activity (e.g. Marienthal et al., 2015; Conlan et al., 2014; Savage, 2006). Conlan and Jamieson (2017) used survey data from avalanche professionals to develop a decision-making tool to aid in forecasting deep slab avalanches. Their tool uses snowpack properties, weather history, and recent avalanche activity to correctly predict deep slab avalanche activity for 77 out of 103 days-roughly 75% of the days used in their study.

A large amount of variability in surface meteorology of the western U.S., and subsequently snow stratigraphy, is driven by synoptic-scale upper atmospheric circulation patterns. Upper-level air movement is a major driver of uplift, which promotes cloud formation and precipitation, and subsidence, which facilitates drier conditions (Barry and Chorley, 2003). Airmass trajectories play an important role in orographic precipitation processes, with the largest precipitation events occurring when a mountain range is oriented perpendicular to the direction of the upper-level flow (Birkeland et al., 2001). Winter precipitation over the western U.S. reveals spatially heterogeneous responses between northern and southern regions, with positive precipitation anomalies in one region coinciding with negative anomalies

in the other. This phenomenon is driven by semi-permanent upper-level circulation patterns, which tend to favor precipitation over either northern or southern regions, but seldom both (Dettinger et al., 1998; Wise, 2010).

The field of synoptic climatology relates synoptic-scale upper atmospheric circulation to the surface environment, including characterizing avalanche activity in the western U.S. (Birkeland et al., 2001), the Selkirk Mountains of British Columbia, Canada (Fitzharris, 1987), Norway (Fitzharris and Bakkehøi, 1986), Iceland (Keylock, 2003), and the Presidential Range in the northeast U.S. (Martin and Germain, 2017). These studies identify significant avalanche events based on magnitude, frequency, destruction, or fatalities. However, there is no research to date relating upper atmosphere circulation patterns specifically to slab avalanches failing on deep persistent weak layers. This work addresses that knowledge gap, with the goal of providing an additional tool for forecasting these difficult-to-predict events through a better understanding of the processes that drive them. This study examines upper air synoptic patterns associated with deep persistent slab avalanching for 3 sites in the western U.S. We identify synoptic circulation types that commonly occur early in the season during years with increased deep persistent slab activity, as well as those that occur commonly in the days immediately prior to deep persistent slab events.

## 2        Study Locations and Methods

### 2.1 Study Locations

This research focuses on three ski areas in the western U.S.: Bridger Bowl, Montana; Jackson Hole, Wyoming; and Mammoth Mountain, California. Bridger Bowl ski area is located on the east side of the Bridger mountain range in southwest Montana approximately 27 km north of Bozeman (45.8174° N, -110.8966° W) with a summit elevation of 2652m and base elevation of 1859m (Figure 1). Jackson Hole Mountain Resort is located in western Wyoming in the Teton Range (43.833° N, -111.871° W). The base elevation at Jackson Hole is 1924 m and the summit rises to 3185 m. Grand Targhee ski area is located in the same mountain range approximately 20 km (12 mi) northwest of Jackson Hole. Due to their close proximity, the records for both of these ski areas are used to describe avalanche activity in the Jackson area for this analysis. Mammoth Mountain Ski Area is located in the Sierra Nevada range in central California (37.630° N, -119.050° W) with a summit elevation of 3698 m and a base elevation of 2424 m. Bridger Bowl and Jackson Hole typically fall within the intermountain snow climate regime (Mock and Birkeland, 2000), characterized by cool winter temperatures and moderate to heavy seasonal snowfall. Mammoth Mountain lies within the coastal snow climate zone, with high seasonal snowfall totals, relatively warm temperatures, and somewhat regular rainfall events during the winter season (Mock and Birkeland, 2000). We selected these study sites because they cover a broad geographical extent within the western U.S. and have some of the longest and most consistent meteorological and avalanche data available.

### 2.2 Meteorological and Avalanche Records

Professional ski patrollers collect daily meteorological and avalanche data at each study site. This research uses the daily data from November 1 to March 31 for each operational season. The operational season depends on snow

coverage, so November data are somewhat sparse for most seasons, depending on the season's opening date. The data record for each site ranges from 32 to 39 seasons (Table 1). Though data are available for earlier years at each site, we did not use them due to the necessity of aligning the data with atmospheric data (discussed in the next section),

and to reduce apparent state changes in the record due to migration of instrument location and changes in recording practices.

We use ski patrol records as they are virtually the only type of record in the US which has a recording rate that is not dependent on fatalities, injuries, or damage to infrastructure. However, there is a distinct difference between a snowpack within the boundaries of a ski area with an active avalanche mitigation program, and a backcountry

snowpack. Active mitigation programs reduce the risk to clients and infrastructure by intentionally triggering many small avalanches throughout the season, with the goal of avoiding a large avalanche later in the season. A ski resort will also have a higher rate of skier compaction, although the effect of this on stability is not always clear, and there are well-documented incidents of large avalanches failing on heavily compacted slopes (e.g. Marienthal et al., 2012). Although mitigation programs continue to become more effective, deep persistent slab avalanches remain a

challenging problem, and there is not an obvious trend over time towards decreasing (or increasing) frequency of such events (Figure 4).

Individual avalanches are characterized by crown depth, bed surface, avalanche type, and relative (R) size (American Avalanche Association, 2016). In most cases, observers record some, but not all of these values for an avalanche. Additionally, observers cannot always ascertain whether avalanches fail within the old snow versus at the interface

between old and new snow. For this reason, we only use the "old snow" bed surface designation in conjunction with other indicators of deep persistent slabs (e.g. type = "hard slab", crown depth $\geq 0.9$m, and R-size $\geq 4$). When observers designate the bed surface as "ground" it indicates that an avalanche failed deep within the snowpack, even if there is some uncertainty as to exactly which layer failed. We used the recorded avalanche type to separate slab and loose avalanches, as well as wet and dry. The classification between 'wet' and 'dry' avalanches is most commonly assessed

at the start zone, although it may also be based on observations from the track or the deposition area. R-size is based on a visual assessment of size relative to the slide path. U.S. avalanche professionals regularly use this size classification, and we use the scores as recorded except for apparent errors (noted in Appendix B of Schauer (2019)).

### 2.3    Synoptic Data

We obtained synoptic atmospheric data from the National Center for Environmental Prediction/ National Center for

Atmospheric Research (NCEP/NCAR) Reanalysis data set (Kalnay et al., 1996), which includes daily 500mb geopotential height values on a 2.5° x 2.5° grid extending from 20° N to 70° N latitude and 160° E to 60° W longitude. A 1197-cell grid describes the spatial distribution of the atmospheric condition for each day. The spatial extent of this study area is similar to previous synoptic studies (e.g. Mock and Birkeland, 2000; Wise and Dannenberg, 2014). We used daily data for the winter season from November 1 – March 31 to correspond with the meteorological and

avalanche data and to avoid a seasonal signal in the circulation patterns after Yarnal (1993).

## 2.4 Avalanche Classification

We classified deep slab avalanches failing on persistent weak layers based on the recorded bed surface, crown depth, avalanche type, R-size, and 72-hour storm totals. Avalanches with a bed surface recorded as "ground" and crown depth greater than 0.9 m are flagged as deep slab events. We also retained events failing in the old snow if crown depth exceeds 0.9 m and is greater than 150% of the mean crown depth from all avalanches for the day. We further consider events using a dimensionless scaling factor representing the ratio of crown height to 72-hour new snow totals:

$$C = \frac{D}{HN_{72}}, \tag{1}$$

where D is the crown depth and HN72 is the 72-hour new snow total. For each site, we select the value of C corresponding with the 99th percentile of the distribution to indicate a deep persistent slab event. This yields C-values of 4.0 for Bridger Bowl, 4.4 for Jackson Hole, and 3.1 for Mammoth Mountain. By considering the tail of the distribution with the larger values for C, we retain the events where the crown depth is large relative to the new snow depth, which should represent deep slab avalanches failing on persistent weak layers.

For avalanches associated with three or more consecutive days of missing precipitation data, we add another set of deep slab avalanches for which crown depth exceeds 0.9 m, the avalanche is classified as a hard slab, the bed surface is designated as "old snow", and the R-size is greater than or equal to 4.

As a final step, we inspect events identified using our classification criteria manually to ensure that the events retained are in fact deep persistent events. Though it is possible these criteria may omit a small number of ambiguous or smaller magnitude events, we maintain that including such events in our record of deep persistent slab events would increase uncertainty in our analysis and make it difficult to assess atmospheric patterns related specifically to deep persistent slab events.

## 2.5 Deep Slab Activity Index

Each avalanche classified as a deep slab event is scored based on the size of the avalanche using the avalanche activity index (AAI) from Schweizer et al. (1998). We refer to this score as the deep slab activity index (DSAI) since any event not classified as a deep slab avalanche receives a score of 0. DSAI scores are summed over each avalanche season (Nov. 1 – Mar. 31), resulting in a seasonal DSAI score:

$$DSAI_y = \sum_{i=1}^{n_y} 10^{R-3}, \tag{2}$$

where $DSAI_Y$ is the seasonal DSAI score for year y, $R$ is the R-size of event $i$, and $n_y$ is the number of deep slab events occurring during year y. The size weighting scheme for the AAI was developed using the D-size classification that estimates the mass of each event, which increases logarithmically with size (American Avalanche Association, 2016). We applied the AAI index to the R-sizes recorded in our dataset based on previous research suggesting the relative mass of an avalanche also increases logarithmically with R-size (Birkeland and Landry, 2002) (Table 2).

## 2.6    Self-Organizing Maps

A self-organizing map (SOM) is a type of neural network that generates a set of descriptive models, or nodes, from a multidimensional dataset. The distribution of these nodes describes the range of variability across the dataset, and each node summarizes a collection of observations that are objectively determined by the neural network model to be categorically similar (Kohonen, 1998). The SOM provides a clear way of summarizing complex multidimensional data, as the algorithm takes into account the similarity between nodes as they are generated and then displays them on a two-dimensional array such that adjacent nodes are similar, while distant nodes are not. SOMs improve on traditional multivariate methods (i.e. principal components analysis or K-means clustering) because they characterize and display synoptic circulation patterns on a defined spectrum of related circulation types, rather than in terms of several independent groups. By initially generating nodes that collectively represent all of the variability within the dataset, the SOM is less sensitive to changes in the number of synoptic types retained than previously used methods such as rotated principal components analysis (PCA). For instance, increasing the number of synoptic types defined by SOMs may highlight subtle changes in synoptic circulation patterns, while increasing the number of types retained using PCA or various clustering algorithms may generate entirely new, and sometimes unrealistic circulation types (Huth, 1996, Reusch et al., 2005). Recent research utilizes SOMs to classify synoptic atmospheric variability (e.g., Sheridan and Lee, 2011; Smith and Sheridan, 2018; Wise and Dannenberg, 2014). However, there is little work applying SOMs to snow and avalanches. Shandro and Haegeli (2018) used SOMs to characterize avalanche types in western Canada for over 14,000 avalanche advisories in order to better understand typical avalanche problems. Schweizer et al. (1994) incorporated SOM, along with Rule-Based Sytems and expert knowledge, to aid in avalanche forecasting in the Swiss Alps. In this work, we apply the methodology developed by previous synoptic climatology studies to the deep persistent slab avalanche problem.

Each SOM node is characterized by 1197 grid point values on the same 2.5° x 2.5° grid used by the NCEP/NCAR daily 500-mb geopotential height maps. The SOM is implemented in R using the Kohonen package (Wehrens and Buydens, 2007) to generate 20 synoptic types summarizing the atmospheric circulation patterns observed over the study area. We explored various SOM configurations using 9, 12, 15, 20, 25, 35, and 56 node-arrays in order to determine an appropriate number of synoptic types to retain. An optimal configuration minimizes variability among synoptic types assigned to the same node and maximizes the variability between nodes. However, if this was the only goal of SOM optimization, one could simply set the number of nodes equal to the number of observations, which would defeat the purpose of implementing a classification scheme. Since the analysis is both more practical and easier to understand with fewer types retained, an optimal classification scheme has to find a balance between interpretability and over-generalization. We found major reduction in within-group variability as the number of nodes increased from 9 to 20, with incremental improvements thereafter. Thus, we identified an optimal configuration of 20 nodes, which is similar to previous synoptic studies (e.g. Esteban et al., 2005; Kidson, 2000; Schuenemann et al., 2009).

Using the new synoptic classification scheme and the daily meteorological and avalanche record, we quantify distributions of daily maximum and minimum temperature and new snow water equivalence (SWE) for each synoptic type at each of the three study sites.  For each site, we calculate median daily maximum and minimum temperatures

for days assigned to each synoptic type. Similarly, we calculate the 75th percentile for daily SWE totals (P75) for days assigned to each synoptic type at each site. Using these descriptive statistics, we identify the synoptic types in the upper and lower quartiles of all the synoptic types for median daily maximum temperature, median daily minimum temperature, and P75 to characterize the coldest, warmest, wettest, and driest circulation patterns at each site. Additionally, we consider the number of days assigned to each synoptic type in the beginning of the winter season (November-January) when deep persistent weak layers tend to form (Marienthal et al., 2015), and compare differences in the frequency distribution of daily synoptic types for major and minor deep slab years at all three study sites. We then count the number of days assigned to each synoptic type in the three days preceding deep slab avalanche activity to identify any relationships between circulation patterns and onset of deep slab avalanches. For each location, we compare frequency distributions that summarize the number of days assigned to each synoptic type from November to January over all major seasons to all minor seasons on the record. Last, we identify similarities and differences between atmospheric setups prior to wet and dry slab activity.

## 3    Results

### 3.1    Map Pattern Classification

The SOM-generated 500 mb map pattern classification scheme is arranged in an array of 5 rows (labelled 1-5, top to bottom) and 4 columns (labelled A-D, left to right) (Figure 2). The array shows a gradual transition from meridional flow (enhanced north-south movement) in the top rows to zonal flow (direct west-east movement) in the bottom rows. There is also a transition from a negative (reverse) Pacific North American (PNA) phase in the lower left corner to a positive phase in the upper right, following main modes of variability as described by Leathers et al. (1991). An upper-level trough over Hudson Bay (referred to hereafter as the Hudson Bay Trough) in the upper left of the array becomes weaker moving diagonally to the lower right. An upper-level ridge over the eastern Pacific Ocean and extending north to the Aleutian Islands and Eastern Siberia in column A migrates eastward moving to the right across the array, until it lies over North America in column D. The eastward migration of this ridge is coupled with development of an upper-level trough over the Aleutian Islands and a transition from a more northwesterly flow over the continental U.S. in column A to a more west-southwest flow in the patterns in column D.

### 3.2    Meteorological Characteristics associated with the Synoptic Types

#### 3.2.1    Bridger Bowl

The coldest synoptic types (A1, A2, A4, B1, B2, B3, and C3) at Bridger Bowl are characterized by a strong north-to-south trajectory over the northwest U.S. (Table 3, Figure 2 and Figure 3). Median daily maximum temperatures for days assigned to these synoptic types range from -3°C to -9°C, while median daily minimum temperatures are -10°C to -19°C. This is in stark contrast to the warmest synoptic types (B4, C2, C5, D2, and D3) at Bridger Bowl, which have median daily maximum temperatures ranging from 2°C to 4°C and median daily minimum temps from -4°C to

-6.6°C. These warm types feature ridging over the Rocky Mountains and a southerly flow trajectory over Bridger Bowl.

Map patterns for types observing the most frequent precipitation events at Bridger Bowl feature a ridge over the eastern Pacific or west coast of the U.S and some development of the Hudson Bay Trough, which results in localized northwest flow over Bridger Bowl (Table 3, Figure 2 and Figure 3). Type B2 has the second highest P75 total with 13mm, with 61% of the days assigned to this type recording some amount of precipitation, making it the wettest of all synoptic types at Bridger Bowl. The Pacific ridge for type B2 is less pronounced than the other five types with frequent precipitation, which results in a more zonal flow pattern and thus more efficient vapor transport as air masses travel a more direct path inland from the Pacific, crossing fewer orographic barriers. The driest synoptic types at Bridger Bowl are characterized by a moderate to strong ridge over the western U.S., leading to increased subsidence in the region and a southwest flow trajectory over the study site. SWE distributions for patterns D2 and C5 have a low percentage of precipitation days (30% and 39%, respectively) but high values for P75 (11.2mm and 12.5mm, respectively), suggesting that storms occurring during these patterns are infrequent but can be large. The SWE distribution for type A4 is noteworthy because although the percentage of days during which precipitation is recorded is the second highest of all types (59%), the P75 value is among the lowest overall (6mm). This map pattern shows the strongest ridge over the east Pacific, and the most intense northerly flow trajectory.

### 3.2.2    Jackson Hole

Similar to the trends observed at Bridger Bowl, the coldest synoptic types at Jackson Hole (A2, A3, A4, B1, B3, and C3) all exhibit an eastern Pacific ridge, Hudson Bay Trough, and a resulting localized northerly flow trajectory (Table 3, Figure 2 and Figure 3). These synoptic types have median daily maximum temperatures ranging from -8.9°C to -5.6°C, and median minimum temperatures from -15.6°C to -12.6°C. The warmest daily temperature distributions are characterized by a southerly flow direction and moderate to strong ridging over the Rockies. These patterns, including B4, C2, C5, D2, and D3, have median daily maximum temperatures ranging from -2.0°C to -1.7°C, with median daily minimum temperatures ranging from -9.2°C to -6.7°C.

 Types A2, A4, B1, B3, and C4 are characterized by a moderate to strong ridge over the east Pacific and the Hudson Bay Trough record precipitation on 77% to 85% of days (Table 3, Figure 2 and Figure 3). Although most days exhibiting these patterns record precipitation, storm totals are near average compared to the rest of the synoptic types, with P75 values between 8.9mm and 10.3mm. The synoptic types with the largest 24-hour storm totals typically exhibit a more zonal pattern. Types A1 and A3 have P75 values of 18.7mm and 11.4mm, respectively, and show a weak ridge over the pacific coast, which leads to a more northwesterly trajectory. Types C4, B4, and D3 have a slightly different configuration, with a trough over the Gulf of Alaska and zonal flow over eastern Canada, which results in zonal flow slightly out of the southwest over the continental U.S. These patterns have P75 values of 11.4 – 15.2mm. The patterns with the smallest percentage of days recording precipitation (40% - 62%) feature enhanced troughing over the Aleutians and Hudson Bay, and a moderate to strong ridge over the western U.S.

### 3.2.3    Mammoth Mountain

Similarly to Bridger Bowl and Jackson, the coldest synoptic types at Mammoth Mountain (A2, A4, B3, and C3) are characterized by a ridge over the east Pacific, strong Hudson Bay Trough, and resulting in a strong northwest flow
(Table 3, Figure 2 and Figure 3). These patterns have median daily maximum temperatures between -1.7°C and 1.4°C, with median daily minimum temperatures from -8.9°C to -10.6°C. Type D5 is one of the coldest synoptic types at Mammoth, but it is also associated with average temperatures at Jackson Hole and above average temperatures at Bridger Bowl. This map pattern shows a split flow at approximately 45° N, with a weak ridge extending to the north and a trough extending to the south.  Types B2, C1, C5, D2, and D3 feature a southwest flow with moderate to strong
ridging over the coast, resulting in warm daily temperature distributions, with median daily maximum temperatures ranging from 6.7°C to 8.3°C and median daily minimum temperatures between -5.8°C and -2.6°C. For type B2, Jackson and Bridger are situated farther downstream of the ridge, which leads to average or below average temperatures at each site.

Types C4, D4, and A5 see among the highest percentage of days with recorded precipitation (44% - 58%), and the
highest P75 totals (38.3mm to 49.5mm). These types are all characterized by a trough over the Gulf of Alaska, with southwest zonal flow patterns over the western U.S. Types A1 and D3 see a smaller percentage of days with precipitation (39% and 35%, respectively), but record high P75 totals (50.8mm and 41.9mm, respectively). These patterns feature a split flow, with an omega block over the Bering Sea and zonal flow over the continental U.S. At Mammoth Mountain, both types result in southwest flow trajectory. Types B3 and A4 show a more northwesterly
flow trajectory, with a ridge over the east Pacific. Although the percentage of days receiving precipitation is relatively high (58% and 44%, respectively), P75 totals are modest (26.2mm and 23.9mm). For the patterns with the smallest percentage of days receiving precipitation (15%-22%), Mammoth Mountain is situated on the downstream end of an upper-level ridge. This results in blocking and a northwest flow trajectory, and is conducive to increased subsidence. Types B5 and C5 see a slightly higher percentage of days with precipitation (29% and 24%, respectively), but
precipitation totals for days associated with the two patterns is among the lowest of all synoptic types (19.2mm and 20.1mm, respectively). These patterns are again associated with enhanced northwest flow (Type B5) and a blocking ridge (Type C5).

### 3.3    Early Season Patterns for Major and Minor Seasons

At each location, we use scatterplots of seasonal cumulative DSAI score to identify clear breaks that separate seasons
with high scores from the rest of the seasons (Figure 4). These seasons are hereafter referred to as "major seasons". Additionally, we designate any season with a cumulative DSAI score of zero as a "minor season". There is a distinct group of seasons with exceptionally high DSAI scores at each site, indicating these seasons had a particularly large number of events or large magnitude events. The threshold separating major seasons from the rest varies by site, as does the number of years designated as major seasons. These differences can be attributed to the differences in the

snow and avalanche climates, and the respective locations as they relate to the atmospheric circulations. At Bridger Bowl, there are four seasons with DSAI scores equal to or exceeding 284, representing the 90th percentile for annual DSAI scores at that site (Table 4). There are four seasons with a DSAI score of zero at Bridger Bowl. We also include the 2014 and 1997 seasons as minor seasons, which had seasonal scores of 0.01 and 0.1, respectively. At Jackson Hole, we find ten seasons with DSAI scores exceeding 209, representing the 75th percentile at that site. Similar to Bridger Bowl, there were only two seasons (1993 and 2003) with seasonal DSAI scores of zero so we include an additional three seasons with DSAI less than or equal to 0.1 as minor seasons (1999, 2005, and 2016). We find seven seasons at Mammoth Mountain with DSAI scores equal to or greater than 100, which corresponds with the 82nd percentile at that site. We found 18 seasons with a seasonal DSAI score of zero at Mammoth Mountain.

### 3.3.1 Bridger Bowl

At Bridger Bowl, there appears to be a shift towards increased meridional flow during the beginning of major DSAI seasons, marked by the large increase in the number of days assigned to types A1, A2, C1, and D1, and the dramatic decrease in counts for types B4 and D4 (Figure 5). Major seasons also see a decrease in counts for type B1, which is characterized by enhanced northwest flow over Bridger Bowl on the back side of a trough, and is commonly associated with increased precipitation. During minor DSAI seasons, there are a large number of days represented by types D2, B4, and C5. Median daily maximum and minimum temperatures for these types are in the upper quartile of all synoptic types at Bridger Bowl.

### 3.3.2 Jackson Hole

The November-January synoptic type counts during major DSAI years at Jackson Hole indicate high counts for the number of days assigned to patterns D1, C2, and D3, all of which are associated with an enhanced ridge over the western U.S. There is a marked decrease in the number of days assigned to the same types during minor seasons (Figure 5). For most major DSAI seasons, the synoptic types with the largest number of days have a weak to moderate Hudson Bay Trough. Major DSAI seasons also have low counts for types A3 and B4, which exhibit a zonal flow pattern with a westerly or slightly southwesterly trajectory over Jackson. The dominant synoptic types during November- January of minor deep slab years in Jackson Hole feature enhanced zonal flow, with very few days assigned to the synoptic types representing blocking patterns in the upper row of the SOM array. Minor DSAI seasons exhibit high frequencies for synoptic types B5, C5, and D5 which may be attributed to above average temperatures or frequent precipitation observed during those circulation patterns.

### 3.3.3 Mammoth Mountain

There is an increase in frequencies for types C2, C4, and D4 during major DSAI seasons at Mammoth Mountain (Figure 5). Type C2 is characterized by enhanced ridging over the West Coast, while types C4 and D4 are zonal patterns with localized southwest flow. There was not a single deep slab event in 18 of the 35 seasons at Mammoth Mountain with complete meteorological and avalanche records. Minor DSAI seasons have a large number of days

assigned to types D2, A3, and C5, which are associated with direct zonal flow and a slight ridge over the coast of California. None of the minor DSAI seasons had large counts for days associated with type A4 or A5. Type A4 is characterized by a strong ridge over the eastern Pacific, with enhanced northwesterly flow over Mammoth Mountain, while type A5 has a slight ridge further to the west over the Aleutians. Both patterns tend to be associated with cold temperatures at Mammoth Mountain.

**3.4     Atmospheric Condition 72 hours Prior to Deep Slab Activity**

At each site, the total number of avalanches during the study period ranges from approximately 20,000 to just over 40,000, while the number of deep slab events is roughly 2-3 orders of magnitude smaller (Table 5). While the recorded deep slab events are primarily dry slab avalanches, there are a small number of deep persistent events classified as wet slabs at Bridger Bowl and Jackson. Each of the three study sites displays a unique distribution of the number of days assigned to each synoptic type in the 72 hours prior to deep slab activity for dry and wet slabs (Figure 6 and Figure 7). We expect a difference between the synoptic conditions for triggering wet and dry deep slabs due to the different processes required to produce these avalanches. At each site a small group of synoptic types occur much more frequently than the others in the three days leading to deep slab activity. The relative frequencies for each type during the 72 hours preceding deep persistent slab activity differ from the overall relative frequency distribution during the study period, which indicates a unique circulation preference for deep persistent slab avalanches at each study site.

**3.4.1     Dry Slab Events**

Types A3, D4, and D5 are associated with the greatest number of days at Bridger Bowl in the 72 hours prior to dry deep slab events (Figure 6). All three types are zonal patterns with varying degrees of troughing over the Aleutian Islands and slightly different north-south orientations (Figure 2). Type B2 was recorded less than ten times in the days leading to dry deep slab events, while type B5 was never observed. Type B2 is characterized by a strong ridge over the east Pacific and Hudson Bay Trough, which results in strong north-south flow over the western U.S., while type B5 shows strong zonal flow from eastern Siberia all the way across to the Atlantic, with a direct westerly trajectory. Types A1, A2, A3, D3 and D4 dominate the period immediately prior to dry deep slab activity at Jackson Hole (Figure 2 and Figure 6). Patterns A1, A2, and A3 are characterized by zonal flow over the western U.S. coming slightly out of the north, and moderate to strong Hudson Bay Trough. In patterns A1 and D3 there is a split flow, with an omega block over the Aleutians and western Alaska and more zonal flow over the continental U.S. However, the two patterns differ in that pattern D3 also shows a weak upper-level ridge over the Rockies, whereas type A1 does not. This leads to a more southwesterly flow in type D3, whereas the trajectory for type A1 is more directly out of the west, and even slightly northwest over Jackson. Pattern D4 shows more enhanced troughing over the Aleutians and the east Pacific, which results in a zonal pattern over the western U.S. with enhanced southwesterly flow. There are very few days associated with types B5, C5, and D5 during the period leading to dry deep slab avalanches at Jackson Hole, all of which are zonal patterns with mild ridging over Jackson.

The 72-hour period prior to dry deep slab events at Mammoth is dominated by types A4 and C4, which are two very different patterns (Figures 2 and 6). Type A4 has the strongest ridge over the Pacific and towards the Gulf of Alaska, while type C4 shows a trough over the Gulf of Alaska that transitions into a direct zonal pattern moving southward into the Pacific. Type A5 was the third most frequent pattern observed in the three days prior to dry deep slab events at Mammoth. This pattern closely resembles type C4 in the southwest quarter of the study area, especially over the central and west Pacific and the California coast. The lack of synoptic types with more intermediate counts at Mammoth Mtn. may be attributed to the smaller number of dry deep slab events compared to the other sites over the duration of the study period.

### 3.4.2    Wet Slab Events

Of the 314 deep persistent slab events recorded at Bridger Bowl, 27 were classified as wet slab avalanches. These occurred on five different days in the record. Nine of the 293 deep slab avalanches were wet slabs at Jackson on six different days, while there were no deep slab events classified as wet slab avalanches at Mammoth Mountain (Table 5). The synoptic types leading to the days with wet slabs at Bridger Bowl show a large number of days associated with type D4, which was also a frequently observed type during dry deep slab events (Figures 6 and 7). The second most common pattern was type C4, which was not as pronounced in the dry deep slab record. Types D3, C4, and D4 were assigned the highest number of days prior to deep wet slab avalanches at Jackson. For both Bridger Bowl and Jackson, the synoptic types assigned the most days prior to deep wet slab activity are characterized by a distinct trough over the Gulf of Alaska, which is coupled with enhanced southwesterly zonal flow over the continental U.S.

## 4        Discussion

### 4.1        Processes related to Deep Persistent Slab Avalanches

The SOM-generated 500 mb maps (Figure 2) provide insight into the key atmospheric processes for persistent deep slab avalanches, both for the early season conditions to generate a snowpack conducive for subsequent avalanches, and for the 72-hour period preceding an event. The predominant synoptic types are different for each of the three locations considered (Figure 6), a function of the location relative to the Pacific Ocean, latitude, altitude, regional topography and site-specific conditions.

This research supports an extensive body of previous work identifying the influence of early-season snowfall and temperatures in the development of snowpack conducive to deep persistent slab avalanches (e.g. Bradley, 1970; Marienthal et al., 2015). Major deep slab seasons at Bridger Bowl and Jackson Hole are characterized by a large number of days with synoptic types associated with low frequency of precipitation events in the beginning of the season. Some of these dry patterns are controlled by a blocking ridge over the continental U.S. (e.g. types C1, D1, and C2 in Fig. 2), which are assigned to a large number of days during major deep slab seasons at both locations and are much less common during minor seasons. Major deep slab seasons at both sites tend to shift towards a positive PNA

pattern, with a high concentration of synoptic types in the upper right corner of the SOM array. This is consistent with research showing these patterns are associated with reduced snowfall totals in the western U.S. (Wise, 2012), and is supported by meteorological records at Bridger and Jackson (Figure 3). Although the difference in 24-hour precipitation (as approximated by the P75 values) varies only by a few millimeters of water among the synoptic types, the difference does have practical significance. For example, the median P75 value for the 5 wettest synoptic types at Bridger Bowl (12.5 mm) is a 134% increase relative to the 5 driest types (9.5 mm). At Jackson hole, the difference is 185%, and at Mammoth Mountain, 210%. As these daily trends are sustained over a period of several weeks and months, these differences will have a strong influence on snowpack structure. Mock and Birkeland (2000) showed that low snowfall totals in the beginning of the season lead to persistent weak layer development by increasing the bulk temperature gradient in intermountain and continental snowpacks.

On the other hand, mild early winter conditions inhibit the formation of weak basal snow layers. This is confirmed by our research, which shows an increase in synoptic types commonly associated with mild temperatures during the beginning of minor seasons (Figure 5). Warmer early season temperatures reduce temperature gradients within the snowpack, thereby minimizing the development of depth hoar or near-surface faceted layers.

A large number of days with synoptic types characterized by cold temperatures (e.g. types A2 and A4) or infrequent precipitation (type C2) occur during November-January of major deep persistent slab seasons at Mammoth Mountain (Figure 5). This is consistent with Bridger Bowl and Jackson Hole. However, unlike the other two study sites, at Mammoth Mountain there are also a substantial number of days in November-January of major seasons assigned to synoptic types that are associated with frequent or heavy precipitation (e.g. patterns A4, D4, and A5). Additionally, there are a large number of days assigned to the drier synoptic types (D2, C3, and B5) during the beginning of minor seasons. Given the processes driving dry snow metamorphism, one would expect the exact opposite- less frequent precipitation leading to enhanced faceting and depth hoar during major seasons (e.g. Armstrong, 1977; Bradley et al., 1977), and increased precipitation with less faceting during minor seasons. However, the majority of the deep persistent slabs at Mammoth Mountain occur in December and January, so sizable precipitation events early in the season are needed to build the thick slabs necessary for deep slab avalanches. Additionally, large precipitation events provide the load required to push a deep persistent weak layer to its breaking point. Indeed, type A4 is associated with regular precipitation at Mammoth Mountain and it shows up frequently during the beginning of major seasons as well as in the 72 hours preceding deep persistent slab events (Figures 5 and 6).

In addition to the large number of days assigned to wet synoptic types in November-January of major seasons at Mammoth, there are also major seasons with high counts for types C2 and C5. These patterns tend to have above-freezing daily maximum temperatures with subfreezing minimum temps. An early season with intermittent precipitation and alternating warm and cold temperatures would facilitate near-surface faceting and ice lens formation, which could act as a low-friction bed surface conducive to avalanches later in the season. Conlan et al. (2014) took field measurements at 41 deep persistent slab avalanches in western Canada, and identified melt-freeze crusts directly below or directly above the failure layer in multiple events. Jamieson et al. (2001) studied deep persistent avalanche cycles for two consecutive seasons in western Canada, both of which were attributed to faceted snow on top of a crust.

This may explain another process by which November-January circulation can favor precipitation at Mammoth Mountain during major deep persistent slab avalanche seasons, especially when combined with periods warm enough to facilitate melt-freeze cycles at the surface, or with rain events early in the season.

During minor deep slab seasons at Mammoth Mountain there simply may not be enough snow to develop a snowpack conducive to deep persistent slab avalanches. This was the case during the 1988 and 1989 seasons, which were dominated by northwest zonal flow (type A3) and the 1991 and 1993 seasons, which had a persistent blocking ridge over the west coast (types C1, C2, and D2). There is also a subset of minor seasons with little to no precipitation during the beginning of the season, but subsequent heavy precipitation for the rest of the season (1992 and 1999). In the coastal climate of Mammoth Mountain, even if there was a persistent weak layer, it may have simply been buried so deeply that no load applied at the surface would be able to initiate a fracture in the weak layer. For the major seasons of 1987, 1996, 1997, and 2001, there was an initial dry period followed by intermediate snowfall. This provided a period during which a persistent weak layer could develop, and enough snow to subsequently bury the persistent weak layer deep enough to form a deep persistent slab, but shallow enough that it remained prone to large applied loads. For these four seasons, all of the deep slab events occurred from November-January.

At all three sites, the 72 hours prior to dry deep slab events are commonly associated with synoptic types characterized by high levels of precipitation (Figure 6). Again, these patterns differ by site depending on latitude, proximity to the coast, and local and regional topography. At Bridger Bowl, the types occurring most commonly during this time period have either enhanced zonal flow (types D4 and D5) or localized northwesterly storm track (type A3). The relatively high counts for types A3 and D4 mimic the frequency pattern for the entire study period. This suggests that these two types show up frequently in this 72-hour window due to their high frequency of occurrence overall, rather than an association with deep persistent avalanches. This is not the case with type D5, which has a very high frequency of occurrence preceding deep persistent slab avalanches despite exhibiting relatively low counts for the duration of the study period. The types with a northwesterly storm track dominate the period prior to deep slabs at Jackson as well; however, there are also very high counts for type D3, which is characterized by a strong southwesterly storm track that channels warm, moist air inland up the Snake River Plain. There is also a distinct spike in patterns A1, A2, and D3 during the 72 hours preceding deep slab activity at Jackson, which stands out when compared to the relatively low counts for all three patterns during the overall duration of the study period. This suggests that these three patterns may indeed be good indicators for increasing likelihood of deep persistent slab avalanches at Jackson Hole given a conducive snowpack. The patterns with the highest count for this 72-hour period at Mammoth (types A4 and C4) exhibit localized west-southwesterly flow (Figures 2 and 6). However, the two patterns differ greatly over the full extent of the study area, with a strong ridge over the Pacific in type A4, and a mild trough in the same area for type C4. Both patterns have frequently recorded high levels for 24-hour precipitation, albeit somewhat less frequently with type A4 than type C4. This may be one reason why type A4 is more frequent than type C4 in the days leading to deep slab avalanches. Similarly to Bridger and Jackson, the synoptic types for the 72-hour period preceding deep persistent slab avalanches at Mammoth look very different than the overall frequency distribution (Figure 6). There is a notable

lack of days associated with patterns C2 and D2 at both sites during the 72 hours prior to dry deep persistent slab avalanches, which are both characterized by a strong blocking ridge that extends over all three study sites (Figure 2). In the days leading to deep slab events, Bridger Bowl records a large number of days with both wetter synoptic types and warmer synoptic types, with a strong southwest flow and a weak ridge over the western U.S. This supports two

mechanisms leading to deep slab events: 1) A large enough load in the form of new snow pushes a deeply buried weak layer to its breaking point in a dry deep slab avalanche, and 2) Rapid warming introduces liquid water to the snowpack thereby causing the weak layer to fail in a wet deep slab avalanche. The latter case provides an explanation for the large number of days assigned to relatively warm synoptic types not commonly associated with precipitation. By isolating the wet slab events from the record, we identify a large number of days assigned to types C4 and D4 at

Bridger Bowl and Jackson Hole in the 72 hours leading to the event (Figure 7). These patterns are characterized by zonal flow with a slight southwesterly component, which results in warm temperatures and frequent precipitation at both sites, although the precipitation totals at Bridger Bowl are usually somewhat modest during these circulation patterns. Jackson Hole also shows a high frequency of type D3, which has a stronger southwest component, usually resulting in even warmer temperatures than types C4 and D4.

The time leading up to wet slab events at both Bridger Bowl and Jackson Hole are not associated with atmospheric ridging over the Gulf of Alaska. This is particularly interesting because types A1 and A2 are among the types with the highest counts for dry deep slab events at Jackson Hole. For these types, a ridge over the Gulf of Alaska is coupled with a northwesterly flow over the continental U.S., often resulting in cold temperatures at both Jackson and Bridger Bowl. Clearly, atmospheric conditions leading up to dry deep slabs differ considerably from conditions leading up to

wet deep slabs. Interestingly, there are no wet slab avalanches on record at Mammoth Mountain that met our criteria for deep slab events. This is somewhat surprising considering Mammoth Mountain is the only location of our three study sites with a maritime snowpack, which is characterized by warmer temperatures and occasional rain-on-snow events throughout the season (Mock and Birkeland, 2000). This lack of deep wet slab events on record may result from a lack of depth hoar development in the maritime snowpack. It may also be that wet slab events occur later in

the season, after the resort has closed and avalanches are no longer being recorded.

At Bridger Bowl, there were six days with deep persistent avalanches that were preceded by rain within the previous 72 hours. In total, there were 9 days with rain recorded to this timeframe. Four of the 9 days were assigned to type D4, 2 were assigned to type C4, and 1 each to types D3, D5, and A3. With the exception of type A3, all of these patterns are characterized by southwesterly zonal flow, while type A3 has a more northwesterly trajectory. These are

515 also the 5 most frequently occurring patterns in the 72 hours prior to deep slab activity for the entire record at Bridger Bowl. There was no rainfall recorded in the 72 hours prior to deep slab activity at Jackson Hole or Mammoth Mountain.

At Bridger Bowl, the 'high-risk' synoptic types (A3, D4, D5, and C4) were observed a total of 230 days in the 72 hours preceding deep persistent slab avalanches (wet and dry). Over the 5 major seasons, these synoptic types were

520 counted 21 times without any subsequent deep slab activity. For Jackson Hole, types A1, A2, A3, C4, D3 and D4 were observed 313 times in the 72 hours prior to deep slab avalanche activity. These types were counted 99 times without

activity over the 10 major seasons. At Mammoth Mountain, types A4 and C4 were counted a total of 65 times in the 72 hours prior to deep slab activity, and 16 times without any subsequent activity over the 7 major seasons. This means that on average, during a season that is prone to deep persistent slab avalanches, these high-risk types occurred an average of roughly 4 days with no activity at Bridger Bowl, 10 days with no activity at Jackson Hole, and 2 days at Mammoth. Although this 'false-positive' rate does not justify using synoptic patterns as the sole predictor of deep slab activity, it does suggest a level of effectiveness that could help resolve some uncertainty associated with predicting these events, when incorporated with other existing forecasting methods.

## 4.2 Application outside the U.S.

While this research focuses on weather and avalanches in the western U.S., our findings are applicable to other mountain regions with similar avalanche problems around the world. Previous research has identified and investigated deep persistent slab avalanches in many other areas around the world, including western Canada (Conlan et al., 2014), the Japanese Alps (Ikeda et al., 2009), the Tien Shan range between western China and Kyrgyzstan (Weilin and Ruji, 1990), and Svalbard (Hancock et al., 2018). While all of these regions may experience deep persistent slab avalanches, our study demonstrates that the events leading to avalanche release will vary with snow climate and many geographic characteristics including continentality, latitude, and local-to-regional topography. However, there are some key points from this research that may be applied at any location. Our findings support previous work that identifies common meteorological factors conducive to deep persistent slab avalanches, specifically low snowfall and cold temperatures in the beginning of the winter season. We used the principles developed in previous synoptic climatology studies to identify circulation patterns that are associated with these meteorological factors, as well as the patterns that commonly occur immediately prior to activity. By identifying these relationships, we are able to clearly identify associations between synoptic-scale atmospheric circulation and deep persistent slab avalanche activity. Thus, we provide a replicable study framework for any site with meteorological and avalanche records, along with a conceptual model that is applicable to many regions around the world exposed to similar hazards.

## 5 Conclusions

This work extends previous synoptic climatology studies linking the atmosphere to avalanche cycles (e.g. Hatchett et al., 2017; Birkeland et al., 2001; Fitzharris, 1987; Martin and Germain, 2017; Schuenemann et al., 2009). We build off of these studies to identify the synoptic drivers of deep persistent slab avalanches, an especially problematic type of avalanche in the western U.S.

We used the state-of-the-art methodology of Self Organized Maps (SOMs) to classify 5,899 daily 500 mb geopotential height maps and generate a continuum of 20 general map patterns that captures major modes of variability over the Pacific, North America, and the West Atlantic. The SOM classification scheme captures different phases of the Pacific-North American teleconnection, Hudson Bay trough, North Pacific circulation centers in the Aleutians and

Siberia, and broad zonal/meridional flow. The array of these four prominent features illustrates a spectrum of synoptic types that that are important for the avalanche sites in the western USA. This work adds to the small body of previous research employing SOM in the field of snow and avalanches, and will hopefully encourage future researchers in the field to make use of machine learning when dealing with large datasets.

        We examined avalanche records at three different locations in the western U.S., and identified atmospheric patterns
that tend to occur at higher rates during years with the most deep slab avalanche activity. Early season patterns tended to be associated with colder temperatures and low snowfall at the two intermountain sites, while the coastal site had some types that are associated with frequent precipitation and others that do not. All three locations had large counts of synoptic types with high SWE frequencies in the 72 hours leading to deep slab avalanches. This supports previous work that identifies recent loading as a leading indicator for deep persistent slab avalanches. Furthermore, the
frequency distribution for the 72-hour period preceding deep persistent slab avalanches at each of the three study sites is unique and distinctly different from the overall distribution during the duration of the study period. This suggests that a vulnerable snowpack at each study site reacts to a unique set of atmospheric circulation patterns. This is promising in terms of avalanche forecasting, as these higher-risk circulation patterns may indicate increasing likelihood of deep persistent slab avalanches. Future research can build off of this work by incorporating numerical
models as another forecasting aid.

        This work improves our understanding of, and our ability to anticipate, deep slab avalanche cycles by highlighting the atmospheric processes driving them. A large amount of uncertainty associated with meteorological and avalanche forecasts exists, and forecasters improve their accuracy by incorporating more pieces of information. This research can improve our understanding of the processes that form a snowpack conducive to deep slab avalanches, and it may
be used as another tool for anticipating when a lurking persistent weak layer might become susceptible to triggering large avalanches. For practitioners working in these three study locations, we provide useful insight into the processes driving deep persistent instability. Many forecasters incorporate synoptic-scale charts in their workflow, as these models tend to be more reliable in the medium term (3-7 days) than mesoscale models, which perform best for forecasts up to 48 hours into the future. They are also useful for improving short-term precipitation prediction (e.g.
Birkeland and Mock, 1996; Birkeland et al., 2001). This work describes associations that may be applied while analyzing synoptic forecast products to provide more lead time in anticipating deep persistent slab avalanche cycles. For those working outside of this study area, this work serves as a replicable framework to draw similar associations at any location with weather and avalanche records. Additionally, the atmospheric classification scheme generated in this study is publicly available (Schauer, 2020) and may be applied to a variety of environmental hazards outside of
the world of snow and avalanches. With continued advancements in understanding the underlying contributing factors associated with deep persistent slab avalanches, we might improve avalanche forecasts, thereby reducing backcountry accidents and economic losses due to infrastructure damage.

## 6        Data Availability

Weather and avalanche records were provided through ski patrol under the strict agreement that the data would not be shared outside of this research group, due to liability concerns per their organizational policies. NCEP Reanalysis data is provided by the NOAA/OAR/ESRL PSD, Boulder, Colorado, USA, from their Web site at https://www.esrl.noaa.gov/psd/. The underlying data for the synoptic classification scheme derived in this study may be obtained at https://doi.org/10.5061/dryad.12jm63xw6.

## 7        Author Contributions

Andrew Schauer developed the methodology, conducted the formal analysis, and managed the data curation under the supervision of Jordy Hendrikx and Karl Birkeland. Andrew Schauer wrote the original draft, which Jordy Hendrikx, Karl Birkeland, and Cary Mock reviewed and edited, providing critical comments and revisions.

## 8        Competing Interests

The authors declare that they have no conflict of interest.

## 9        Acknowledgments

We wish to thank Pete Maleski, Doug Richmond, Richard Griffen, and Ella Darham of Bridger Bowl Ski Patrol for sharing meteorological and avalanche data and discussing its use and limitations. Ned Bair and Mammoth Mountain Ski Patrol provided records for Mammoth Mountain, and Chris McCollister, Bob Comey, and Patrick Wright of the
Bridger-Teton National Forest Avalanche Center provided records for the Jackson area, which were maintained by Jackson Hole and Grand Targhee ski patrols. M.D. Higgs contributed countless hours of editing and feedback in the analysis.  Funding was provided through the student research grant from the American Avalanche Association.

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

**11 Figures**

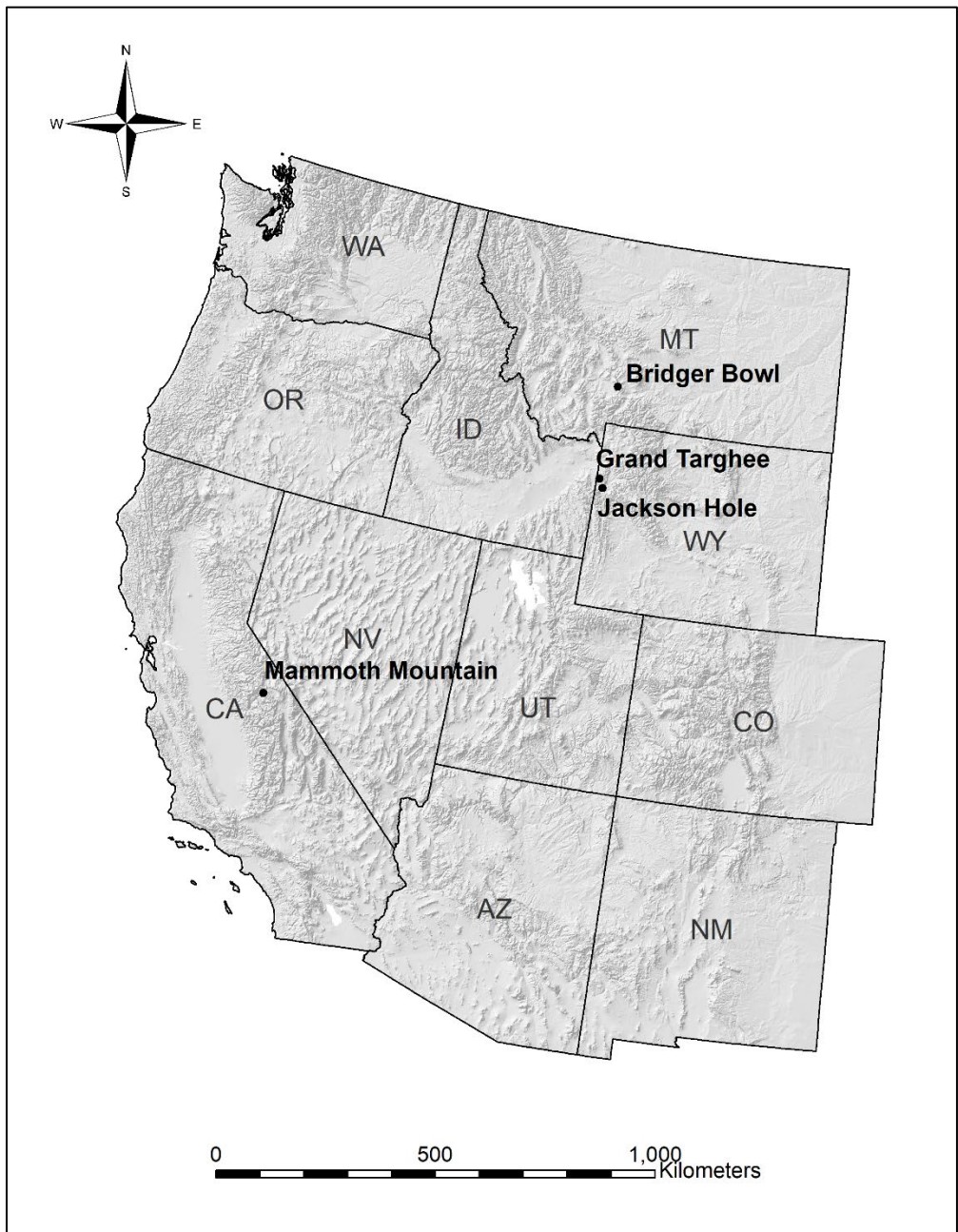

**Figure 1: Location map of the sites used in this study. Geographic data from US Census Bureau (2017) and USGS National Center for EROS (2005).**

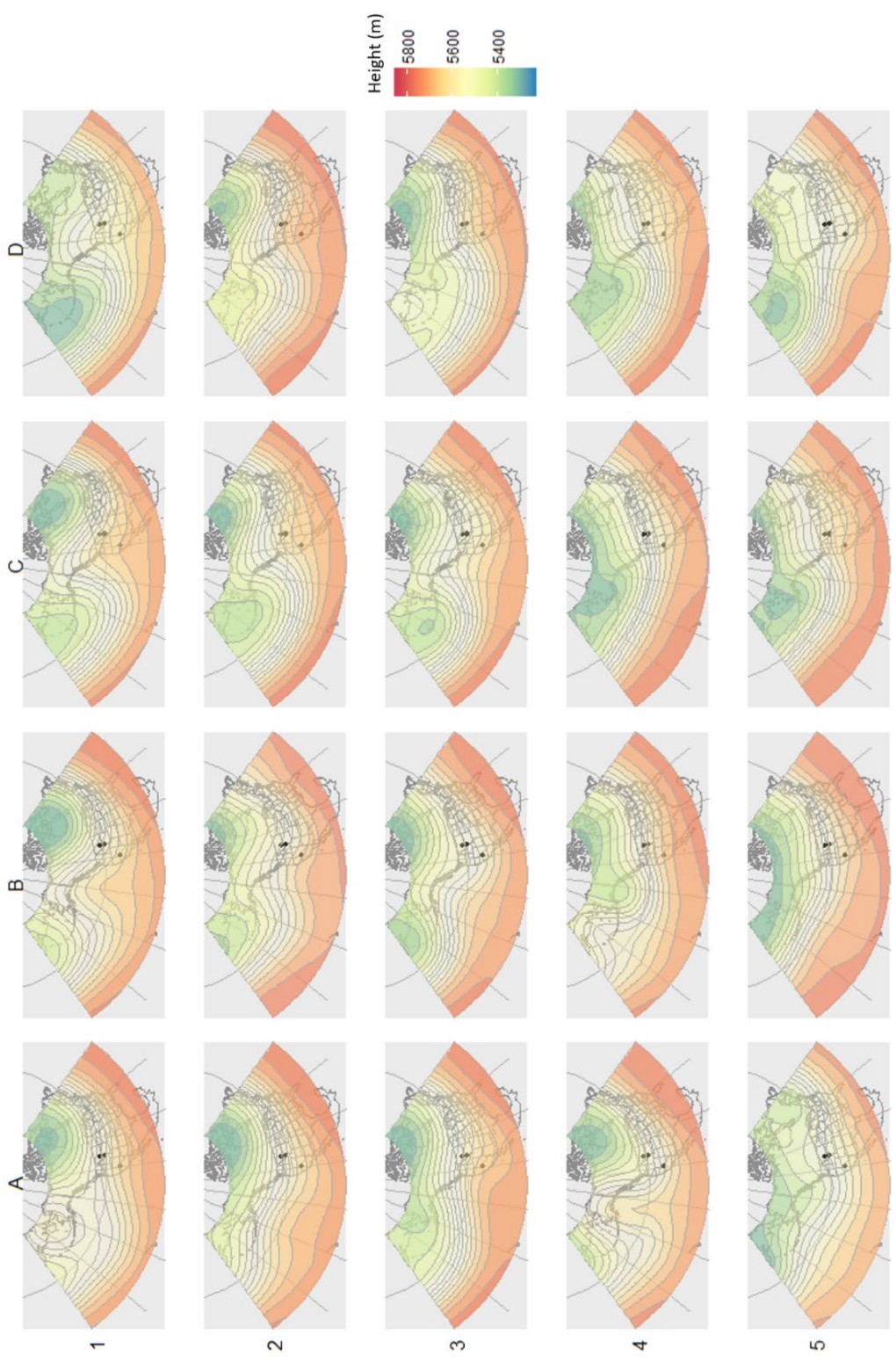

**Figure 2: Array of 500 mb geopotential height map patterns generated using self-organizing maps. Each map pattern, or synoptic type, is a generalization of a group of days exhibiting a similar circulation pattern.**

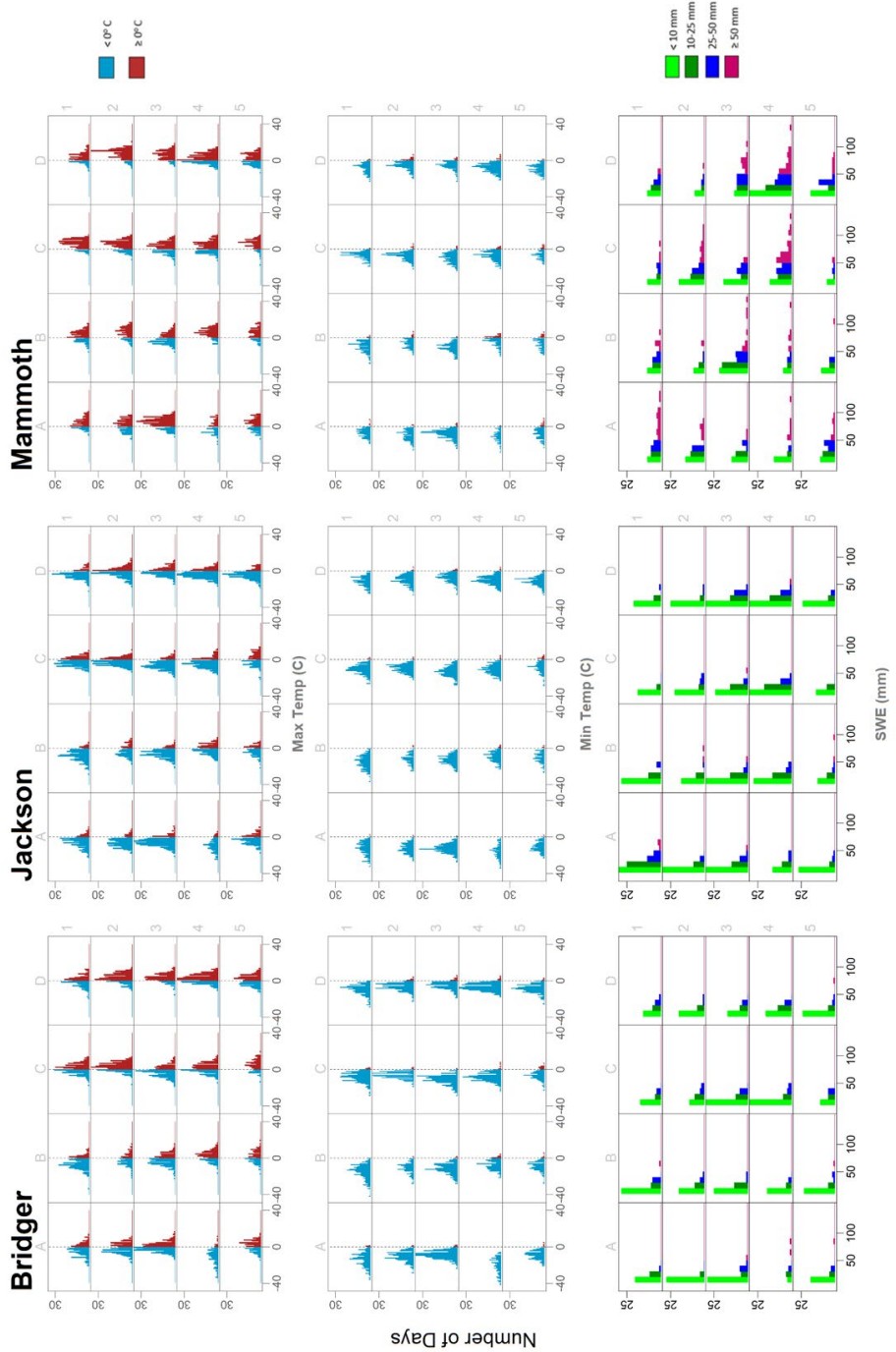

Figure 3: Histograms of daily maximum temperature (top row), minimum temperature (middle row), and snow water equivalence totals (bottom row) for Bridger Bowl (left), Jackson (center), and Mammoth Mountain (right). For each of the nine arrays, the position of the plot corresponds with the map pattern shown in Figures 2 and 3. For example, the histograms located in the upper left corner of each array correspond with synoptic type A1.

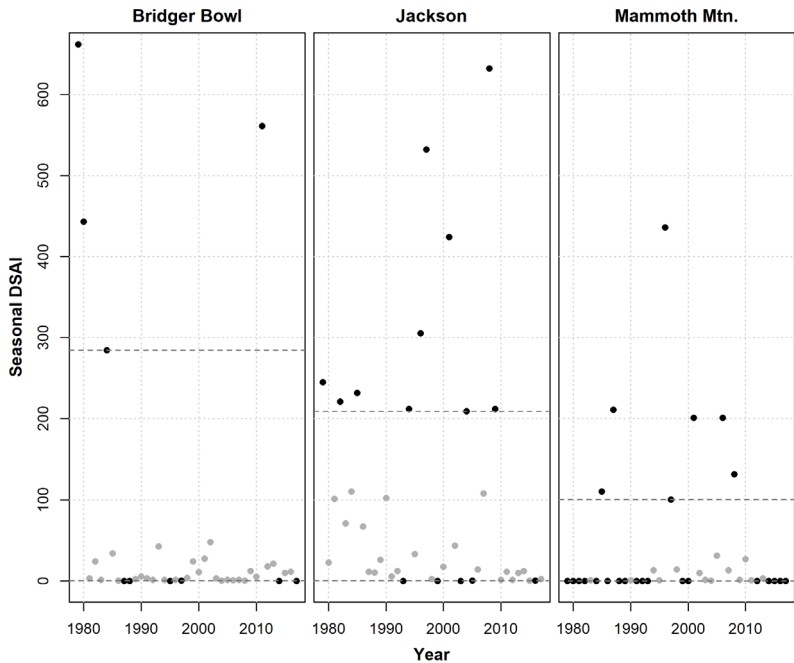

**Figure 4: Scatterplots of seasonal deep slab activity index scores for Bridger Bowl, MT (left), Jackson Hole (center), and Mammoth Mountain, CA (right). Highlighted points represent the years considered as major and minor deep persistent avalanche seasons, and dashed lines represent the cutoff thresholds used to identify those points.**

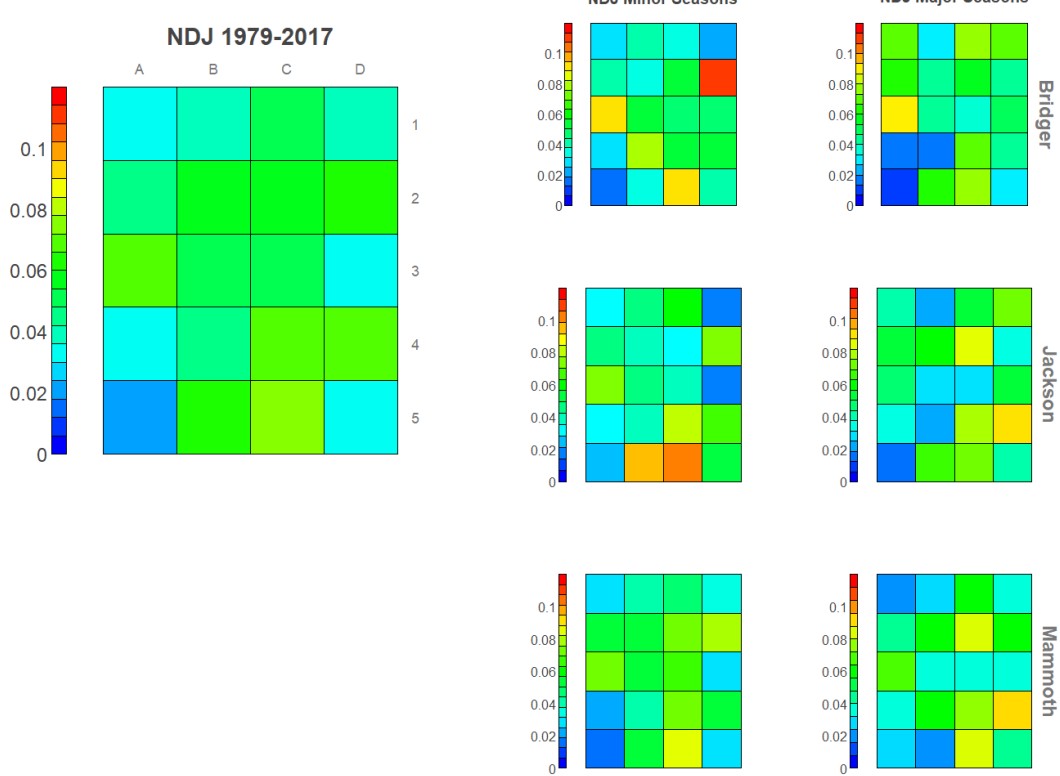

**Figure 5: Relative frequencies for each synoptic type from November-January for the entire study period (upper left), over all minor seasons (left column), and all major seasons (right column). Heat maps are given for Bridger Bowl (top row), Jackson (center row), and Mammoth Mountain (lower row). Frequencies are calculated by summing counts of days assigned to each synoptic type over all major (minor) seasons and dividing by total number of days for all major (minor) seasons.**

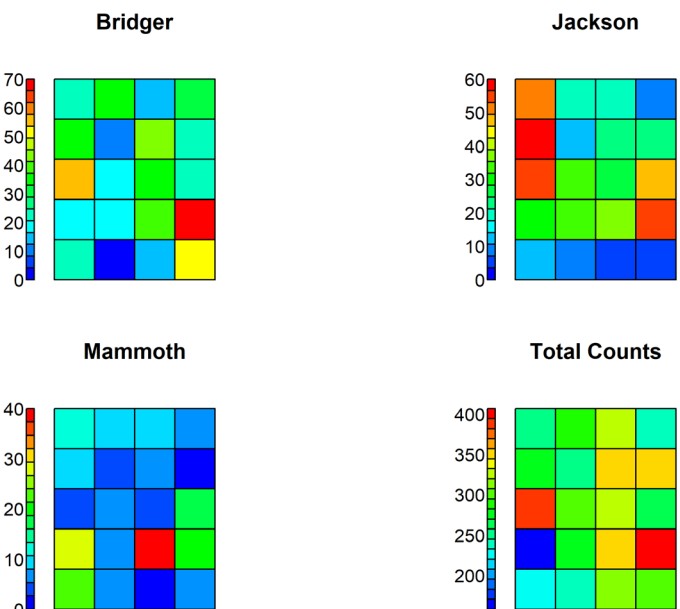

**Figure 6: Relative frequencies for each synoptic type within 72 hours prior to a day with dry deep slab avalanches recorded at Bridger Bowl (upper left), Jackson Hole (upper right), Mammoth Mountain (lower left), and total counts for the duration of the study period.**

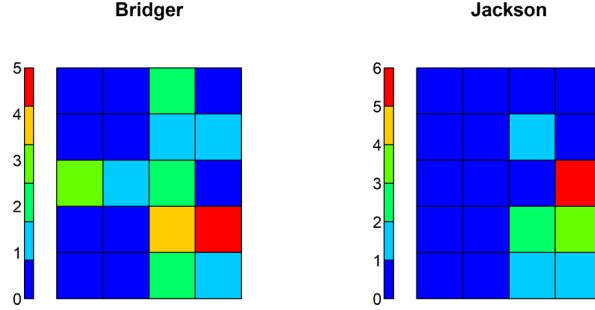

**Figure 7: Counts for each synoptic type within the 72 hours prior to deep wet slab avalanches at Bridger Bowl (left) and the Jackson area (right).**

## 12    Tables

**Table 1: Length of avalanche and meteorological records used at each study site.**

| Location | Avalanche Record | Meteorological Record |
|---|---|---|
| Bridger Bowl | 1979/80 – 2017/18 | 1979/80 - 2017/18 |
| Jackson Hole | 1979/80 - 2017/18 | 1979/80 - 2017/18 |
| Mammoth Mountain | 1979/80 - 2017/18 | 1982/83 - 2013/14 |

**Table 2*: DSAI score for each R-size designation.**

| Avalanche R-size | DSAI Score |
|---|---|
| 1 | .01 |
| 2 | .1 |
| 3 | 1 |
| 4 | 10 |
| 5 | 100 |

820

Table 3: Descriptive statistics for meteorological metrics at all three locations. SWE is summarized with the 75th percentile of daily SWE for all days assigned to each synoptic type, along with percent of total days during which any amount of precipitation was recorded. Daily maximum and minimum temperatures are summarized using median values for all days assigned to each synoptic type. For each column, the top five wettest or warmest types are highlighted in bold, while the five driest or coldest types are italicized.

| | Bridger | | | | | Jackson | | | | Mammoth | | | |
|---|---|---|---|---|---|---|---|---|---|---|---|---|---|
| | Total # Days | Min T (C) Median | Max T (C) Median | SWE (mm) % Days >0 | SWE (mm) P75 | Min T (C) Median | Max T (C) Median | SWE (mm) % Days >0 | SWE (mm) P75 | Min T (C) Median | Max T (C) Median | SWE (mm) % Days >0 | SWE (mm) P75 |
| **A1** | 256 | *-12* | -3 | 53 | *10* | -11.1 | -4.4 | 75 | **18.7** | -6.7 | 2.8 | 39 | **50.8** |
| **B1** | 288 | *-13* | *-6* | **59** | 10.8 | *-14.4* | *-7.8* | **77** | 8.5 | *-8.3* | 2.2 | 29 | 33 |
| **C1** | 332 | -7 | 0 | *32* | *9.5* | -10 | -2.8 | *40* | *6.4* | **-5.6** | **6.7** | *21* | *19.9* |
| **D1** | 244 | -9 | -1 | 46 | *9.3* | -11.1 | -4.4 | 64 | *6.4* | -6.7 | 2.8 | 36 | 23.6 |
| **A2** | 272 | -10 | *-3* | 43 | **12** | *-12.8* | *-6.1* | **78** | 10.3 | *-8.9* | *1.4* | 35 | **34.3** |
| **B2** | 254 | -10 | *-3* | **61** | **13** | -11.7 | -3.9 | 68 | 8.8 | -5.8 | **6.1** | *20* | 23.2 |
| **C2** | 356 | **-6** | **2** | *29* | 10 | **-9.2** | **-2.2** | 53 | *6.7* | -6.7 | 5 | 37 | 30.5 |
| **D2** | 350 | **-6** | **2** | *30* | 11.4 | **-8.9** | **-1.7** | *44* | 7.6 | **-5** | **8.3** | *15* | *18.4* |
| **A3** | 391 | -8 | -1 | 41 | 11 | *-12.8* | -5.6 | 69 | **11.4** | -6.7 | 4.7 | *19* | 21.6 |
| **B3** | 302 | *-11* | -2 | **58** | 10 | -12.2 | -5.6 | **82** | 8.9 | *-10.6* | *-1.1* | **58** | 26.2 |
| **C3** | 329 | -10 | *-3* | **55** | 11 | *-13.3* | *-6.1* | 76 | 7.6 | -8.9 | 2.2 | 36 | *20.4* |
| **D3** | 258 | **-4** | **3** | 42 | *9.3* | **-6.7** | **-1.7** | 68 | **15.2** | **-4.4** | **6.7** | 35 | **41.9** |
| **A4** | 158 | *-19* | *-9* | **59** | *6* | *-15.6* | *-8.9* | **85** | 8.9 | *-10* | *-1.7* | **44** | 23.9 |
| **B4** | 277 | **-6.6** | **2** | 43 | 11 | **-8.9** | **-1.7** | 62 | **12.4** | **-2.8** | 5.6 | 22 | 37.1 |
| **C4** | 355 | -8 | 0 | 50 | 11 | -10.6 | -4.4 | **79** | **11.4** | -6.7 | 2.2 | **48** | **49.5** |
| **D4** | 407 | -7 | 1 | *37* | 10 | -10 | -3.9 | 71 | 10.2 | -7.2 | *1.7* | **49** | **38.5** |
| **A5** | 223 | -8 | 1 | 50 | 11 | -11.1 | -2.8 | 75 | 8.9 | -6.7 | 3.3 | **47** | **38.3** |
| **B5** | 237 | -8 | 1 | 52 | **15.1** | -10 | -3.1 | 65 | 10.9 | **-5.6** | 4.4 | 29 | *19.2* |
| **C5** | 312 | **-6** | **4** | *39* | **12.5** | **-7.8** | **-0.6** | 53 | 8 | **-5.6** | **6.1** | 24 | *20.1* |
| **D5** | 298 | -8 | 0 | 54 | 10 | -11.1 | -3.9 | 72 | *6.4* | *-8.9* | *1.1* | 37 | 34 |

**Table 4: Major and minor deep slab avalanche seasons for the three study sites, determined by cumulative seasonal DSAI score.**

| Location | Major Seasons | Minor Seasons |
|---|---|---|
| Bridger Bowl | 1979, 1980, 1984, 2011 | 1987, 1988, 1995, 1997, 2014, 2017 |
| Jackson Hole | 1979, 1982, 1985, 1994, 1996, 1997, 2001, 2004, 2008, 2009 | 1993, 1999, 2003, 2005, 2016 |
| Mammoth Mountain | 1985, 1987, 1996, 1997, 2001, 2006, 2008 | 1979-1982, 1984, 1986, 1988, 1989, 1991-1993, 1999, 2000, 2012, 2014-2017 |

**Table 5: Summary of number of total avalanches, all deep slab events, dry deep slab events, and wet deep slab events. The number of days each type of avalanche was recorded is given in parentheses.**

| | All Avalanches | All Deep Slab | Dry Deep Slab | Wet Deep Slab |
|---|---|---|---|---|
| Bridger Bowl | 31455 (2161) | 314 (176) | 287 (169) | 27 (7) |
| Jackson Hole | 20180 (2343) | 293 (173) | 284 (168) | 9 (5) |
| Mammoth Mountain | 41751 (1177) | 92 (60) | 92 (60) | 0 (0) |