# Peer review of "Synoptic atmospheric circulation patterns associated with deep persistent slab avalanches in the western United States"

_Natural Hazards and Earth System Sciences, 2020_

## Referee Comment (RC1) · Anonymous Referee #1 · 16 Nov 2020

The study builds on previous research and covers a novel approach to relate large scale weather patterns in early winter season combined with recent snowfall amounts to major and minor avalanche cycles with avalanches fracturing deep in the snow cover. These cycles are generally difficult to foresee and can cause considerable damage and loss of human lives. The study covers different climatic regions in western US and long data series. The analysis takes avalanche activity data with snow cover characteristics into consideration. The study is well organized and the paper well written. The results are valuable for the community in that it adds knowledge on the relationship of climato-logic snow cover characteristics to avalanche activity. To underline the strength of the method and to put it into a forecasting perspective, I recommend to show and discuss

in more detail the cases, when long- and short-term weather patterns are typical for deep slab avalanche cycles, but the cycle does not occur as well as on the advantages over a pure correlation with 72h storm totals. Furthermore, please consider are a few more small comments below.

Introduction: Line 34: Please cite literature on avalanche formation in persistent weak layers for wet avalanches. Line 57: Please also consider the work by Sturm, Holmgren et al. (1995) Climatic snowpack classification system; with classes for the seasonal snowcover according to stratigraphic and textural attributes.

Study locations and methods: All study sites are ski resort, explain in more detail how disturbed/undisturbed the snow cover is by skiers and explosives, and how this has changed over time and how this may influence avalanche activity and avalanche size. Please show your definition of wet versus dry avalanche (where is the snow wet? - in the starting zone and/or path and/or deposition zone). How it it recorded? Deep wet avalanches are often correlated with first wetting of deep layers. Is this data available? Avalanche classification: Avalanche dynamic studies show, that the crown-depth but also the amount of erodable snow (new and old snow) in avalanche paths determines the dynamics and runout length of large avalanches. Hence, it is important, from where the measurements of the 72h storm totals are. Is there a difference in altitude of the location the measurement and the avalanche crown and/or the avalanche path? Please explain in more detail.

Discussion: Please discuss the reverse case in more detail, where typical long- and short-term weather patterns for deep slab cycles are present, but no minor or major deep slab avalanche cycles occurred. Please discuss the advantage of the presented method over simple correlation with 72h storm totals.

line 405: There is also subset -> There is also a subset line 436: wetter -> weather

---

## Referee Comment (RC2) · Anonymous Referee #2 · 23 Nov 2020

General Comments

This is a well-written manuscript that utilizes a novel technique to assess synoptic-scale patterns that may promote deep persistent slab avalanche activity. The results may be useful for avalanche practitioners and researchers in U.S. close to the study sites and perhaps elsewhere. General comments are as follows:

1. You do not describe the snowpack setup conducive for a deep persistent slab avalanche; only minor discussion of depth hoar and faceting is provided in the discussion. I suggest including such information, as it is paramount to have such a snowpack setup for a deep persistent slab avalanche to occur.

2. The introduction does not refer to the relatively large amount of research conducted on deep persistent slab avalanches. A more thorough literature review would help set the layout of where this research fits into the overall goal of predicting such events.

3. I'd appreciate more details in describing the usefulness of this researcher for an avalanche forecaster, practitioner, or researcher. When may someone use this approach rather than assessing the snowpack to see if it is prime for such an avalanche type? I find it difficult to think that someone would assess the synoptic-scale weather rather than the snowpack to forecast for such avalanches. How can someone implement your findings within their overall workflow to better predict these hard-to-predict events?

4. Please discuss the potential for false alarms. Given these synoptic flow types, how many days did not experience deep persistent slab avalanche activity?

5. The SWE P75 values in Table 3 do not appear substantially different for the different flow types; for example, the differences are often single digits of mm SWE. Please better describe in the manuscript why this may be the case and if there are thresholds that a practitioner or researcher could use to better predict the release of deep persistent slab avalanches.

6. Consider comparing your parameter values to other studies. Are the precipitation and warming values comparable to other studies or substantially different, and why do you suspect this to be the case?

Specific Comments

Line 32: This suggests that no persistent deep slab avalanches release due to snowpack warming, or that they must be wet, i.e., due to melting. Consider reviewing.

Line 73: What snow climate are Bridger and Teton in? This is already listed for Mammoth.

Line 111: In terms of confidence in the dataset, it could be helpful to speak to whether

each avalanche was confirmed to have released on the recorded day or if interpretation/expert judgement was used to identify time of release.

Line 113: Why was 72-hour storm total used in this analysis? Please indicate its usefulness over 24-hour, 48-hour, or longer storm totals.

Line 127: What criteria was used to manually ensure each case was a deep slab? Were there comments associated with the avalanches indicating so, or information that they failed on deeply buried weak layers?

Line 194: Some of the top rows, for example 1D looks to have meridional flow. Are you referring to specifically over your study sites? If so, worth clarifying.

Line 381: I suspect any avalanche forecaster/practitioner would be able to tell you this. I suspect there are substantially more references that indicate a low early-season snow year can lead to late-season persistent deep slab avalanches. Perhaps worth deliberately stating that this paragraph is not a new finding but agrees with extensive previous observations and research.

Line 400: Perhaps some of the precipitation discussed occurred as rain, forming a melt-freeze crust. Jamieson et al. (2001) found a facet-on-crust snowpack setup particularly prone to deep persistent slab avalanche activity. This again refers to the manuscript not discussing snowpack setup prone to such avalanches.

Line 412: Some deep persistent slab avalanches occurred in November? Would these weak layers not only be a couple of weeks old?

Line 438: Perhaps also increased strain rate?

Figure 1: International residents may not know the state codes. Consider writing out state names, and perhaps in a darker colour for improved legibility.

Table 3: Maximum daily air temperatures appear to be quite high. As an avalanche practitioner, I would certainly be wary of a snowpack with a deeply buried weak layer

and a daily high air temperature close to or above 0°C. Even more so if such warm air temperatures were prolonged over days. Can this be further described/refined? Further, how many of these above-freezing air temperature days correspond to rain vs. no precipitation and does this correlate with activity?

Table 4: Bridger Bowl and Jackson Hole appear relatively similar in terms of location in Figure 1. It is interesting that Bridger Bowl and Jackson Hole seasons hardly align. Please describe why this may be the case, or if this is expected.

Technical Corrections

Line 138: period missing after y

Line 198: A should be An

Line 288: Move the 'a' to before 'seasonal'

---

## Author Comment (AC1) · 2 Jan 2021

We would like to extend our sincere thanks to the editor Yves Bühler and the two anonymous reviewers for taking the time to review and comment on our manuscript. Attached you will find our responses to the general and specific comments provided to us. Note that we refer to line numbers referencing the modified and resubmitted document. Thanks again for your consideration and help bringing this manuscript to publication!

Please also note the supplement to this comment:

[Figure]

https://nhess.copernicus.org/preprints/nhess-2020-302/nhess-2020-302-AC1-supplement.pdf

[Figure]

**Supplement:**

**Response to review comments**

**Manuscript Number:**         NHES-2020-302

**Submitted title:**         **Synoptic atmospheric circulation patterns associated with deep persistent slab avalanches in the western United States**

*We would like to extend our sincere thanks to the editor Yves Bühler and the two anonymous reviewers for taking the time to review and comment on our manuscript. Below you will find our responses to the general and specific comments provided to us. Note that we refer to line numbers referencing the modified and resubmitted document. Thanks again for your consideration and help bringing this manuscript to publication!*

Comments from the editors and reviewers: (in plain text)

**Responses from the co-authors (bold)**

**Reviewer #1 Responses**

*General Comments*

The study builds on previous research and covers a novel approach to relate large scale weather patterns in early winter season combined with recent snowfall amounts to major and minor avalanche cycles with avalanches fracturing deep in the snow cover. These cycles are generally difficult to foresee and can cause considerable damage and loss of human lives. The study covers different climatic regions in western US and long data series. The analysis takes avalanche activity data with snow cover characteristics into consideration. The study is well organized and the paper well written. The results are valuable for the community in that it adds knowledge on the relationship of climatologic snow cover characteristics to avalanche activity. To underline the strength of the method and to put it into a forecasting perspective, I recommend to show and discuss in more detail the cases, when long- and short-term weather patterns are typical for deep slab avalanche cycles, but the cycle does not occur as well as on the advantages over a pure correlation with 72h storm totals. Furthermore, please consider are a few more small comments below

**Thank you for these comments. We have provided additional details in the discussion as to how this method is of value for forecasting, and some additional information on how this reduces uncertainty for these types of events, over a 72hr storm total. The specific details are listed below. We have also addressed all of your comments below:**

Introduction: Line 34: Please cite literature on avalanche formation in persistent weak layers for wet avalanches.

**We added references to Baggi and Schweizer (2009), Marienthal et al. (2012), and Pietzsch (2009) to line 36 in the updated manuscript.**

Line 57: Please also consider the work by Sturm, Holmgren et al. (1995) Climatic snowpack classification system; with classes for the seasonal snowcover according to stratigraphic and textural attributes

**This paragraph is included to provide background relevant to synoptic climatology, which explicitly links atmospheric circulation to surface properties. The Sturm, Holmgren, and Liston (1995) snow cover classification paper is more closely related to the Mock and Birkeland (2000) paper, which uses meteorological variables (e.g. temperature, precipitation, rain events) to characterize different snowpacks, and does not fit into the synoptic climatology framework. The snowpack climatology classification system is relevant to this study, and we reference the Mock and Birkeland (2000) paper since it is more recent and more applicable to our study area (line 106 in the updated manuscript).**

Study locations and methods: All study sites are ski resort, explain in more detail how disturbed/undisturbed the snow cover is by skiers and explosives, and how this has changed over time and how this may influence avalanche activity and avalanche size. Please show your definition of wet versus dry avalanche (where is the snow wet? – in the starting zone and/or path and/or deposition zone). How it it recorded? Deep wet avalanches are often correlated with first wetting of deep layers. Is this data available? Avalanche classification: Avalanche dynamic studies show, that the crown-depth but also the amount of erodable snow (new and old snow) in avalanche paths determines the dynamics and runout length of large avalanches. Hence, it is important, from where the measurements of the 72h storm totals are. Is there a difference in altitude of the location the measurement and the avalanche crown and/or the avalanche path? Please explain in more detail.

**We added a more thorough description of the ski area record to the methods section (lines 116-126 in the revised manuscript).**

**We included a sentence with regards to the wet vs. dry designation (line 134 in the revised version). These records are used as part of a holistic approach to assess snowpack stability from an operational perspective. They can be recorder-dependent and do not necessarily involve direct measurements or a thorough examination of the avalanche. However, for reasons mentioned previously, they remain the most reliable and consistent record available in the US. Our classification process (described in section 2.4) minimizes the effect of this uncertainty by taking multiple metrics into account to identify large avalanches.**

Discussion: Please discuss the reverse case in more detail, where typical long- and short-term weather patterns for deep slab cycles are present, but no minor or major deep slab avalanche cycles occurred. Please discuss the advantage of the presented method over simple correlation with 72h storm totals

**We added a section on this the discussion (lines 518-528).**

**This method is not intended to replace any existing methods (i.e. weather, snowpack, and avalanche observations, test results, etc.). However, the tools we currently have available often leave a large amount of uncertainty in predicting the timing of these events. Our results should be able to reduce that uncertainty by providing another indicator of increasing likelihood. We added text (lines 576-585) with respect to incorporating our findings in a typical workflow already used by practitioners.**

line 405: There is also subset -> There is also a subset line 436: wetter -> weather

**Line 405 error fixed. Line 436 is intended to say wetter, referring to patterns with more precipitation.**

We trust that the editor agrees that we have considered, and generally agreed with the vast majority of the peer review and editor comments provided for our submission. These review comments have greatly helped improve the clarity of the paper, and the narrative provided.

Kind regards

Andrew Schauer

(on behalf of the author team)

**References used in this response:**

Baggi, S., and Schweizer, J.: Characteristics of wet=snow avalanche activity: 20 years of observations from a high alpine valley (Dischma, Switzerland), Nat. Haz., 50, 97-108, doi: 10.1007/s11069-008-9322-7, 2009.

Marienthal, A., Hendrikx, J., Birkeland, K.W., and Irvine, K.: Meteorological variables to aid forecasting deep slab avalanches on persistent weak layers, Cold Reg. Sci. Technol., 120, 227-236, doi:10.1016/j.coldregions.2015.08.007, 2015.

Mock C.J., and Birkeland, K.W.: Snow Avalanche Climatology of the Western United States Mountain Ranges, Bull. Am. Meteorol. Soc., 87 (10), 2367-2392, doi: 10.1175\/1520-0477(2000)081<2367:sacotw>2.3.co;2, 2000.

Pietzsch, E.H.: Water movement in a stratified and inclined snowpack: Implications for wet slab avalanches, MSc thesis, Montana State University, Bozeman, MT, USA, 2009.

Sturm, M., Holmgren, J., and Liston, G.E.: A seasonal snow cover classification system for local to global applications, J. Clim., 8 (5), 1261-1283, doi: https://doi.org/10.1175/1520-0442(1995)008%3C1261:ASSCCS%3E2.0.CO;2, 1995.

---

## Author Comment (AC2) · 2 Jan 2021

We would like to extend our sincere thanks to the editor Yves Bühler and the two anonymous reviewers for taking the time to review and comment on our manuscript. Attached you will find our responses to the general and specific comments provided to us, as well as one revised figure (Fig. 01). Note that we refer to line numbers referencing the modified and resubmitted document. Thanks again for your consideration and help bringing this manuscript to publication!

Please also note the supplement to this comment:

[Figure]

https://nhess.copernicus.org/preprints/nhess-2020-302/nhess-2020-302-AC2-supplement.pdf

[Figure]

[Figure]

**Fig. 1.**

**Supplement:**

**Response to review comments**

| | |
|---|---|
| **Manuscript Number:** | **NHES-2020-302** |
| **Submitted title:** | **Synoptic atmospheric circulation patterns associated with deep persistent slab avalanches in the western United States** |

*We would like to extend our sincere thanks to the editor Yves Bühler and the two anonymous reviewers for taking the time to review and comment on our manuscript. Below you will find our responses to the general and specific comments provided to us. Note that we refer to line numbers referencing the modified and resubmitted document. Thanks again for your consideration and help bringing this manuscript to publication!*

Comments from the editors and reviewers: (in plain text)

**Responses from the co-authors (bold)**

**Reviewer #2 Responses**

*General Comments*

This is a well-written manuscript that utilizes a novel technique to assess synoptic-scale patterns that may promote deep persistent slab avalanche activity. The results may be useful for avalanche practitioners and researchers in U.S. close to the study sites and perhaps elsewhere. General comments are as follows:

**Thank you for these comments. We have provided specific responses to each of your comments below:**

1. You do not describe the snowpack setup conducive for a deep persistent slab avalanche; only minor discussion of depth hoar and faceting is provided in the discussion. I suggest including such information, as it is paramount to have such a snowpack setup for a deep persistent slab avalanche to occur.

**We added additional text (lines 38-53) to the Introduction describing the structure and mechanics of a snowpack conducive to deep slab avalanches. As this is already a lengthy manuscript, we have tried to keep the introduction as brief as possible, while still providing the background and references necessary to provide context for this research.**

2. The introduction does not refer to the relatively large amount of research conducted on deep persistent slab avalanches. A more thorough literature review would help set the layout of where this research fits into the overall goal of predicting such events.

**Again, we have deliberately kept the introduction and literature review brief in the interest of keeping the manuscript at a reasonable length. We also need to balance space given to climatology with that dedicated to snow and avalanches. That said, your point on highlighting previous research relevant to this topic is well-taken. We added an additional paragraph (lines 63-71) highlighting previous work pertaining to meteorological variables related to deep persistent slab avalanches, which complements the previous**

**paragraphs on deep slab characteristics and mechanics (e.g. Marienthal et al., 2015; Conlan et al., 2014; Savage, 2006).**

3. I'd appreciate more details in describing the usefulness of this researcher for an avalanche forecaster, practitioner, or researcher. When may someone use this approach rather than assessing the snowpack to see if it is prime for such an avalanche type? I find it difficult to think that someone would assess the synoptic-scale weather rather than the snowpack to forecast for such avalanches. How can someone implement your findings within their overall workflow to better predict these hard-to-predict events?

**This research is not intended to replace existing methods currently in use by practitioners; rather, we explicitly state our intention of providing more information to complement those existing tools (e.g. lines 27, 63, and 503-506 in the preprint). Current methods available to practitioners are well-equipped to address avalanches failing in the mid- and upper snowpack, but leave a large amount of uncertainty when applied to deep persistent slab avalanches. This uncertainty lies in the fact that deep persistent weak layers can exist for days, weeks, or months with little to no avalanche activity before becoming reactive (e.g. Statham et al., 2018; Marienthal et al., 2012), and that field tests used to approximate stability are poorly suited to assess the likelihood of a human triggering an avalanche on a deep persistent weak layer. We have added more text in the conclusion (lines 576-585 in the revised version) to describe the application of this work to sites located within our study area, beyond our study area, as well as to problems unrelated to snow and avalanches.**

4. Please discuss the potential for false alarms. Given these synoptic flow types, how many days did not experience deep persistent slab avalanche activity?

**We added a section on this the discussion (lines 518-528 in the revised version).**

5. The SWE P75 values in Table 3 do not appear substantially different for the different flow types; for example, the differences are often single digits of mm SWE. Please better describe in the manuscript why this may be the case and if there are thresholds that a practitioner or researcher could use to better predict the release of deep persistent slab avalanches.

**We have added a brief discussion of this topic to the manuscript (lines 418-422 in the revised version).**

6. Consider comparing your parameter values to other studies. Are the precipitation and warming values comparable to other studies or substantially different, and why do you suspect this to be the case?

**Although we use meteorological metrics to characterize each synoptic pattern, we do not attempt to identify precipitation or warming thresholds as has been done in previous studies (e.g. Conlan and Jamieson, 2017; Conlan and Jamieson, 2016; Marienthal et al., 2015). The nature of a synoptic climatology study is to identify large-scale patterns to describe a process, rather than small-scale metrics to parameterize thresholds.**

*Specific Comments*

Line 32: This suggests that no persistent deep slab avalanches release due to snowpack warming, or that they must be wet, i.e., due to melting. Consider reviewing.

**Our review of the literature, as well as the lengthy avalanche forecasting experience of some of those in our team of authors, suggest that there is little direct evidence for deep slab avalanches releasing only due to warming and without the addition of liquid water to the snowpack from either rain or melting. In particular,**

Schweizer and Jamieson (2010) state that "Whereas measurements have shown that the surface layers in fact creep more rapidly due to warming, field evidence is mostly lacking on how these changes affect snow instability. This might be because the effects of surface warming are subtle and/or only observable under certain slab/weak layer conditions". In our experience these conditions are exceedingly rare, whereas cases of deep wet slab avalanches due to liquid water in the snowpack are much more common (e.g., Kattleman, 1984; Peitzsch et al., 2010).

Conlan et al. (2014) identified warming trends prior to 20 out of 36 avalanches deep persistent slab avalanches, but noted that "rapid warming or cooling is less important for release but trends could be indicative of other processes such as storm events". Reuter and Schweizer (2012) found "Critical cut lengths tended to decrease with decreasing slab stiffness, suggesting that surface warming increases crack propagation propensity", but go on to note that the change in slab stiffness due to warming would not have a strong enough effect on its own to cause a slab to release: "However, the effect seems to be subtle. It is suggested that a pre-existing weakness and significant energy input are required for surface warming to promote instability". It is also worth noting these conclusions were made with respect to a weak layer buried 40 cm deep; deeper weak layers would experience an even smaller effect from surface warming.

Although it may be possible, there is relatively little literature documenting deep persistent slab failure due to warming-induced changes in strain rates. Thus, with regards to warming, we only mention the case with liquid water in our brief introduction.

Line 73: What snow climate are Bridger and Teton in? This is already listed for Mammoth.

See lines 79-82 in the preprint, or lines 102-104 in the revised version.

Line 111: In terms of confidence in the dataset, it could be helpful to speak to whether each avalanche was confirmed to have released on the recorded day or if interpretation/ expert judgement was used to identify time of release

Our avalanche records were maintained by ski patrol at each of the ski resorts used in this study. The events are recorded daily, so it is unlikely that any of the large avalanches we are interested in would go unnoticed on a given day. There were a few days when a ski area had to close because avalanche danger was so high. In these rare cases, the timing of the avalanches was subject to expert judgment and recorded within the comments. We therefore have a high level of confidence in the timing of these events.

Line 113: Why was 72-hour storm total used in this analysis? Please indicate its usefulness over 24-hour, 48-hour, or longer storm totals.
This was a conservative approach to avoid including direct-action avalanches that failed as a result of new storm snow. Statham et al. (2018) define storm slab avalanches as failing within 'a few hours or days' after a storm. In the absence of a strict definition characterizing the time frame limiting storm snow avalanches, we relied on previous professional experience that it would be highly unlikely that a storm snow avalanche would occur beyond three days after a storm. It may well be that it would have been equally valid to use a 24- or 48-hour, or longer window to isolate these events. It is because of uncertainties such as these that we checked the comments attached to each avalanche event to avoid designating direct-action avalanches as deep persistent slabs.

Line 127: What criteria was used to manually ensure each case was a deep slab? Were there comments associated with the avalanches indicating so, or information that they failed on deeply buried weak layers?

There were comments associated with these avalanches. Since the full record contained over 90,000 avalanches, it was not feasible to read the comments for the full record. We automated our classification using the methods described in section 2.4 to reduce that dataset, and checked the comments for the records identified with our algorithm in order to remove events that clearly failed within new snow, at the interface between new and old snow, or at a weak layer near the surface of the snowpack.

Line 194: Some of the top rows, for example 1D looks to have meridional flow. Are you referring to specifically over your study sites? If so, worth clarifying.

**Here we refer to regional patterns. The arrangement of the SOM array is such that meridional patterns are located in the top rows of the array, trending towards zonal patterns near the bottom. It follows that pattern 1D, being in the top row, should resemble a meridional pattern.**

Line 381: I suspect any avalanche forecaster/practitioner would be able to tell you this. I suspect there are substantially more references that indicate a low early-season snow year can lead to late-season persistent deep slab avalanches. Perhaps worth deliberately stating that this paragraph is not a new finding but agrees with extensive previous observations and research

**We re-wrote the first sentence of this paragraph (line 408 in the revised manuscript) to better emphasize the extensive work already describing this process.**

Line 400: Perhaps some of the precipitation discussed occurred as rain, forming a meltfreeze crust. Jamieson et al. (2001) found a facet-on-crust snowpack setup particularly prone to deep persistent slab avalanche activity. This again refers to the manuscript not discussing snowpack setup prone to such avalanches.

**We added a short paragraph (lines 442-455 in the revised version) discussing the role buried crusts might have played in deep slab cycles at Mammoth Mountain.**

Line 412: Some deep persistent slab avalanches occurred in November? Would these weak layers not only be a couple of weeks old?

**Although uncommon, it is not unheard of to observe deep persistent slab avalanches this early in the season. This was actually the case in the Chugach National Forest this year, with deep persistent slab avalanches first mentioned in the backcountry avalanche advisory on Nov. 28. Conlan and Jamieson (2016) studied 88 deep persistent slab avalanches and found the weak layers to be as fresh as 16 days old.**

**Another factor influencing this observation is that our avalanche data is recorded through ski patrol avalanche mitigation programs. During the middle of the season, patrol is able to perform mitigation daily to trigger many smaller avalanches in order to avoid the exact events we are studying in this paper. It is not unexpected that deep slab avalanches would be triggered by explosives early in the season, when control work is just beginning, and deep persistent weak layers may not yet have experienced any kind of loading similar in magnitude to an artillery shot or a hand charge. Again, these events are not the norm- for Mammoth Mountain, deep slab avalanches were recorded on seven days in November over the 38-season record.**

Line 438: Perhaps also increased strain rate?

**See our response to comments regarding line 32 and concepts related to strain rate above.**

Figure 1: International residents may not know the state codes. Consider writing out state names, and perhaps in a darker colour for improved legibility

**We updated figure 1, making the labels a darker font. Using the full state names made the map quite busy. Although not ideal, the state abbreviations are an international standard (ISO 3166-2) and are readily available to those who may not be familiar with these titles.**

Table 3: Maximum daily air temperatures appear to be quite high. As an avalanche practitioner, I would certainly be wary of a snowpack with a deeply buried weak layer and a daily high air temperature close to or above 0_C. Even more so if such warm air temperatures were prolonged over days. Can this be further described/refined? Further,

how many of these above-freezing air temperature days correspond to rain vs. no precipitation and does this correlate with activity?

**While median values for daily maximum temperatures are at or above freezing for many of the synoptic types, all of the daily minimum temperatures are well below freezing, reflecting the unlikely occurrence of prolonged periods with sustained temperatures above freezing. The high daily temperatures at the Mammoth Mountain study site are also a result of the location of their weather station, which is located near the base of the mountain. For Bridger Bowl and Jackson Hole, the weather stations used in their historical records are located mid-elevation in the resorts and temperatures are therefore closer to what might be experienced in start zones at higher elevations. While this is important to note, we do not use these values for anything other than distinguishing relatively warm synoptic types from relatively cold types, or similarly, relatively wet types from relatively dry ones. This issue would be more concerning if we were trying to identify specific meteorological thresholds contributing to deep slab avalanches.**

**Rain was not a substantial driver in terms of avalanche activity at these three sites. We added a brief section discussing the relationship between rain, deep slab activity, and circulation patterns to the discussion (lines 511-517 in the revised version).**

Table 4: Bridger Bowl and Jackson Hole appear relatively similar in terms of location in Figure 1. It is interesting that Bridger Bowl and Jackson Hole seasons hardly align. Please describe why this may be the case, or if this is expected

**It is an interesting note. Although they are closer to each other than to Mammoth Mountain, Bridger and Jackson rarely experience similar snowpacks during a given season. This is a function of the major role regional topography plays in winter-season precipitation in the intermountain west. The concept is addressed in Birkeland et al. (2001), which studied the relationship between upper-level circulation patterns and avalanche activity at four sites in the western U.S., including Bridger Bowl and Jackson Hole. In their conclusion, they state "distinctive atmospheric conditions are associated with avalanche extremes at each site. These differing patterns can be largely explained by the topography of the region, and the locations of the sites in relation to various mountain barriers and low-elevation pathways for moisture".**

**Technical Corrections**

Line 138: period missing after y
Line 198: A should be An
Line 288: Move the 'a' to before 'seasonal'

***All issues addressed as recommended.***

We trust that the editor agrees that we have considered, and generally agreed with the vast majority of the peer review and editor comments provided for our submission. These review comments have greatly helped improve the clarity of the paper, and the narrative provided.

Kind regards

Andrew Schauer

(on behalf of the author team)

**References used in this response:**

Birkeland, K.W., Mock, C.J., and Shinker, J.: Avalanche extremes and atmospheric circulation patterns, Ann. Glaciol., 32, 135-140, doi:10.3189/172756401781819030, 2001.

Conlan, M., and Jamieson, B.: A decision support tool for dry persistent deep slab avalanches for the transitional snow climate of western Canada, Cold Reg. Sci. Technol., 144, 16-27, 2017.

Conlan, M., and Jamieson, B.: Naturally triggered persistent deep slab avalanches in western Canada Part I: avalanche characteristics and weather trends from weather stations, J. Glaciol., 62 (232), 243-255, doi: 10.1017/jog.2016.1, 2016.

Conlan, M., Tracz, D., and Jamieson, B.: Measurements and weather observations at persistent deep slab avalanches, Cold Reg. Sci. and Technol., 97, 104-112, doi:10.1016/j.coldregions.2013.06.011, 2014.

Kattleman, R.:  Wet slab instability.  Proceedings of the 1984 International Snow Science Workshop, 1984.

Marienthal, A., Hendrikx, J., Birkeland, K.W., and Irvine, K.: Meteorological variables to aid forecasting deep slab avalanches on persistent weak layers, Cold Reg. Sci. Technol., 120, 227-236, doi:10.1016/j.coldregions.2015.08.007, 2015.

Marienthal, A., Hendrikx, J., Chabot, D., Maleski, P., and Birkeland, K.W.: Depth Hoar, Avalanches, and Wet Slabs: A case study of the historic March 2012 wet slab avalanche cycle at Bridger Bowl, Montana, Proceedings of the International Snow Science Workshop, Anchorage, AK, USA, 16-21 September, 2012.

Peitzsch, E., Hendrikx, J., and Fagre, D.:  Characterizing wet slab and glide slab avalanche occurrence along the Going-to-the-Sun road, Glacier National Park, Montana, USA, Proceedings of the 2010 International Snow Science Workshop, 2010.

Reuter, B., and Schweizer, J.: The effect of surface warming on slab stiffness and the fracture behavior of snow, Cold Reg. Sci. Technol., 83–84, 30-36. https://doi.org/10.1016/j.coldregions.2012.06.001, 2012.

Schweizer, J. and Jamieson, B.:  On surface warming and snow instability.  Proceedings of the 2010 International Snow Science Workshop, 2010.

Statham, G., Haegeli, P., Greene, E., Birkeland, K., Israelson, C., Tremper, B., Stethem, C., McMahon, B., White, B., Kelly, J.: A conceptual model of avalanche hazard, Nat. Haz., 90, 663-691, doi: https://doi.org/10.1007/s11069-017-3070-5, 2018.